# Compositional Policy Learning in Stochastic Control Systems with Formal Guarantees

**Đorđe Žikelić**[*]
Institute of Science and Technology Austria (ISTA)
Klosterneuburg, Austria
djordje.zikelic@ist.ac.at

**Mathias Lechner**[*]
Massachusetts Institute of Technology
Cambridge, MA, USA
mlechner@mit.edu

**Abhinav Verma**
The Pennsylvania State University
University Park, PA, USA
verma@psu.edu

**Krishnendu Chatterjee**
Institute of Science and Technology Austria (ISTA)
Klosterneuburg, Austria
krishnendu.chatterjee@ist.ac.at

**Thomas A. Henzinger**
Institute of Science and Technology Austria (ISTA)
Klosterneuburg, Austria
tah@ist.ac.at

## Abstract

Reinforcement learning has shown promising results in learning neural network policies for complicated control tasks. However, the lack of formal guarantees about the behavior of such policies remains an impediment to their deployment. We propose a novel method for learning a composition of neural network policies in stochastic environments, along with a formal certificate which guarantees that a specification over the policy's behavior is satisfied with the desired probability. Unlike prior work on verifiable RL, our approach leverages the compositional nature of logical specifications provided in SPECTRL, to learn over graphs of probabilistic reach-avoid specifications. The formal guarantees are provided by learning neural network policies together with reach-avoid supermartingales (RASM) for the graph's sub-tasks and then composing them into a global policy. We also derive a tighter lower bound compared to previous work on the probability of reach-avoidance implied by a RASM, which is required to find a compositional policy with an acceptable probabilistic threshold for complex tasks with multiple edge policies. We implement a prototype of our approach and evaluate it on a Stochastic Nine Rooms environment.

## 1 Introduction

Reinforcement learning (RL) has achieved promising results in a variety of control tasks. However, the main objective of RL is to maximize expected reward [56] which does not provide guarantees of the system's safety. A more recent paradigm in safe RL considers constrained Markov decision processes (cMDPs) [5, 26, 3, 21], that in addition to a reward function are also equipped with a cost function. The goal in solving cMDPs is to maximize expected reward while keeping expected cost below some tolerable threshold. Recently, keeping cost below the threshold almost-surely and not only in expectation was considered in [52]. While these methods do enhance safety, they only empirically

---

[*]Equal contribution.

try to minimize cost and do not provide guarantees on cost constraint satisfaction. The lack of formal guarantees raises a significant barrier to deployment of these methods in safety-critical applications, where the policies often need to encode complicated, long-horizon reasoning to accomplish a task in an environment in which unsafe behavior can lead to catastrophic consequences [6].

Recent work [13, 32, 33] has explored decomposing high-level logical specifications, belonging to different fragments of Linear Temporal Logic (LTL) [44], into simpler logical specification tasks that RL algorithms can solve more easily. In particular, these methods first decompose the high-level logical specification into a number of simpler logical specifications, then learn policies for these simpler tasks and finally compose the learned policies for each task into a global policy that solves the high-level control problem at hand. The simpler control tasks are solved by designing reward functions which faithfully encode the simpler logical specifications obtained by the decomposition and using an off-the-shelf RL algorithm to solve them.

While these methods present significant advances in deploying RL algorithms for solving complex logical specification tasks, their key limitation is that they also do not provide formal guarantees on the probability of satisfying the logical specification which are imperative for safety-critical applications. This is because reward functions often only "approximate" logical specification objectives, and furthermore RL algorithms are in general not guaranteed to converge to optimal policies over continuous state spaces. Recently, [30, 62] have proposed verification procedures for formally verifying a composition of neural network policies with respect to a given system dynamics model. However, these works assume that the underlying control systems have *deterministic dynamics*. This does not allow modelling *stochastic disturbances and uncertainty* in the underlying system dynamics. Furthermore, [62] assumes that the underlying control system has a *linear dynamics function* whereas many control tasks have *non-linear dynamics*.

In this work we propose CLAPS (Compositional Learning for Probabilistic Specifications), a new compositional algorithm for solving high-level logical specification tasks with formal guarantees on the probability of the specification being satisfied. We present a method for learning and verifying a composition of neural network policies learned via RL algorithms, which is applicable to *stochastic control systems* that may be defined via *non-linear dynamics functions*. Our method only requires that the underlying state space of the system is compact (i.e. closed and bounded) and that the dynamics function of the system is Lipschitz continuous. In terms of the expressiveness of logical specifications, CLAPS considers the SPECTRL specification language [32]. SPECTRL contains all logical specifications that can be obtained by sequential and disjunctive compositions of reach-avoid specifications. Given a target and an unsafe region, a reach-avoid specification requires that the system reaches the target region while avoiding the unsafe region.

Our method learns a policy along with a formal certificate which guarantees that a specification is satisfied with the desired probability. It consists of three key ingredients – (1) high-level planning on a directed acyclic graph (DAG) that decomposes the complex logical specification task into sequentially or disjunctively composed low-level reach-avoid tasks, (2) learning policies that solve low-level reach-avoid tasks with formal guarantees on the probability of satisfying reach-avoidance, and (3) composing the low-level reach-avoid policies into a global policy for the high-level task by traversing edges in the DAG. This yields a fully *compositional* algorithm which only requires solving control problems with reach-avoid constraints, while performing high-level planning.

To solve the low-level reach-avoid tasks, we leverage and build upon the recent learning algorithm for stochastic control under reach-avoid constraints [68]. This work achieves formal guarantees on the probability of reach-avoidance by learning a policy together with a reach-avoid supermartingale (RASM), which serves as a formal certificate of probabilistic reach-avoidance. When composing such guarantees for a global probabilistic specification we encounter two fundamental challenges, (i) reducing the certification of unnecessary individual reach-avoid constraints that are not critical for global satisfaction, and (ii) reducing the violation probabilities for individual reach-avoid constraints to improve the possibility of satisfaction of the composed policy. We overcome the first challenge by leveraging the DAG decomposition of complex specifications and by performing a forward pass on the DAG which learns edge policies *on-demand*, and the second one by showing that RASMs prove a *strictly tighter* lower bound on the probability of reach-avoidance than the bound that was originally proved in [68]. Our novel bound multiplies the bound of [68] by an exponential asymptotic decay term. This is particularly important when composing a large number of edge policies for complex

objectives. Furthermore, our new bound is of independent interest and it advances state of the art in stochastic control under reach-avoid constraints.

**Contributions** Our contributions can be summarized as follows:

1. We prove that RASMs imply a strictly tighter lower bound on the probability of reach-avoidance compared to the bound originally proved in [68] (Section 4).
2. We propose a novel method for learning and verifying a composition of neural network policies in *stochastic control systems* learned in the RL paradigm, which provides *formal guarantees* on the probability of satisfaction of SPECTRL specifications (Section 5).
3. We implement a prototype of our approach and evaluate it on the Stochastic Nine Rooms environment, obtained by injecting stochastic disturbances to the environment of [33]. Our experiments demonstrate both the necessity of decomposing high-level logical specifications into simpler control tasks as well as the ability of our algorithm to solve complex specification tasks with formal guarantees (Section 6).

## 2 Related Work

**Learning from logical specifications** Using high-level specifications with RL algorithms has been explored in [13, 32, 33, 39, 43, 60] among others. These approaches typically use the specification to guide the reward objective used by a RL algorithm to learn a policy but usually lack a formal guarantee on the satisfaction of the specification. Approaches that provide guarantees on the probability of satisfaction in finite state MDPs have been proposed in [12, 28, 29]. In contrast, we consider stochastic systems with continuous state spaces.

**Control with reachability and safety guarantees** Several works propose model-based methods for learning control policies that provide *formal guarantees* with respect to reachability, safety and reach-avoid specifications. The methods of [2, 16, 17, 48, 55, 1] consider deterministic systems and learn a policy together with a Lyapunov or barrier function that guarantees reachability or safety. Control of finite-time horizon stochastic systems with reach-avoid guarantees has been considered in [14, 36, 53, 61]. These methods compute a finite-state MDP abstraction of the system and solve the constrained control problem for the MDP. The works of [9, 10] consider infinite-time horizon stochastic systems with linear dynamics and propose abstraction-based methods that provide probably approximately correct (PAC) guarantees. Control of polynomial stochastic systems via stochastic barrier functions and convex optimization was considered in [46, 49, 65, 42]. Learning-based methods providing formal guarantees for infinite time horizon stochastic systems that are not necessarily polynomial have been proposed in [38, 68, 41, 20]. The work [8] proposes a learning-based method for providing formal guarantees on probability 1 stabilization. To the best of our knowledge, the compositionality of our CLAPS makes it the first method that provides formal guarantees for infinite time horizon stochastic systems with respect to more expressive specifications.

**Safe exploration** RL algorithms fundamentally depend on exploration in order to learn high performing actions. A common approach to safe exploration RL is to limit exploration within a high probability safety region which is computed by estimating uncertainty bounds. Existing works achieve this by using Gaussian Processes [11, 34, 59], linearized models [23], robust regression [40], Bayesian neural networks [37], shielding [4, 7, 25, 31] and state augmentation [51].

**Model-free methods for stochastic systems** The recent work of [63] proposed a model-free approach to learning a policy together with a stochastic barrier function, towards enhancing probabilistic safety of the learned policy. Unlike our approach, this method does not formally verify the correctness of the stochastic barrier function hence does not provide any guarantees on safety probability. However, it does not assume the knowledge of the system environment and is applicable in a model-free fashion. Several model-free methods for learning policies and certificate functions in deterministic systems have also been considered, see [24] for a survey.

**Statistical methods** Statistical methods [67] provide an effective approach to estimating the probability of a specification being satisfied, when satisfaction and violation of the specification can be witnessed via finite execution traces. Consequently, such methods often assume a finite-horizon or the existence of a finite threshold such that the infinite-horizon behavior is captured by traces within that threshold. In environments where these assumptions do not hold, the violation of a reach-avoid (or, more generally, any SPECTRL ) specification can be witnessed only via infinite traces and

hence statistical methods are usually not applicable. In this work, we consider *formal verification* of *infinite-time horizon systems.*

**Supermartingales for probabilistic programs** Supermartingales have also been used for the analysis of probabilistic programs (PPs) for properties such as termination [15], reachability [58] and safety [18, 19].

## 3 Preliminaries

We consider a discrete-time stochastic dynamical system whose dynamics are defined by the equation

$$\mathbf{x}_{t+1} = f(\mathbf{x}_t, \pi(\mathbf{x}_t), \omega_t),$$

where $t \in \mathbb{N}_0$ is a time step, $\mathbf{x}_t \in \mathcal{X}$ is a state of the system, $\mathbf{u}_t = \pi(\mathbf{x}_t) \in \mathcal{U}$ is a control action and $\omega_t \in \mathcal{W}$ is a stochastic disturbance vector at the time step $t$. Here, $\mathcal{X} \subseteq \mathbb{R}^n$ is the state space of the system, $\mathcal{U} \subseteq \mathbb{R}^m$ is the action space and $\mathcal{W} \subseteq \mathbb{R}^p$ is the stochastic disturbance space. The system dynamics are defined by the dynamics function $f : \mathcal{X} \times \mathcal{U} \times \mathcal{W} \to \mathcal{X}$, the control policy $\pi : \mathcal{X} \to \mathcal{U}$ and a probability distribution $d$ over $\mathcal{W}$ from which a stochastic disturbance vector is sampled independently at each time step. Together, these define a *stochastic feedback loop system.*

A sequence $(\mathbf{x}_t, \mathbf{u}_t, \omega_t)_{t=0}^{\infty}$ of state-action-disturbance triples is a *trajectory* of the system, if we have that $\mathbf{u}_t = \pi(\mathbf{x}_t)$, $\omega_t \in \mathrm{support}(d)$ and $\mathbf{x}_{t+1} = f(\mathbf{x}_t, \pi(\mathbf{x}_t), \omega_t)$ hold for each time step $t \in \mathbb{N}_0$. For every state $\mathbf{x}_0 \in \mathcal{X}$, we use $\Omega_{\mathbf{x}_0}$ to denote the set of all trajectories that start in the initial state $\mathbf{x}_0$. The Markov decision process (MDP) semantics of the system define a probability space over the set of all trajectories in $\Omega_{\mathbf{x}_0}$ under any fixed policy $\pi$ of the system [47]. We use $\mathbb{P}_{\mathbf{x}_0}$ and $\mathbb{E}_{\mathbf{x}_0}$ to denote the probability measure and the expectation operator in this probability space.

**Assumptions** For system semantics to be mathematically well-defined, we assume that $\mathcal{X} \subseteq \mathbb{R}^n$, $\mathcal{U} \subseteq \mathbb{R}^m$ and $\mathcal{W} \subseteq \mathbb{R}^p$ are all Borel-measurable and that $f : \mathcal{X} \times \mathcal{U} \times \mathcal{W} \to \mathcal{X}$ and $\pi : \mathcal{X} \to \mathcal{U}$ are continuous functions. Furthermore, we assume that $\mathcal{X} \subseteq \mathbb{R}^n$ is compact (i.e. closed and bounded) and that $f$ and $\pi$ are continuous functions. These are very general and standard assumptions in control theory. Since any continuous function on a compact domain is Lipschitz continuous, our assumptions also imply that $f$ and $\pi$ are Lipschitz continuous.

**Probabilistic specifications** Let $\Omega$ denote the set of all trajectories of the system. A *specification* is a boolean function $\phi : \Omega \to \{\text{true}, \text{false}\}$ which for each trajectory specifies whether it satisfies the specification. We write $\rho \models \phi$ whenever a trajectory $\rho$ satisfies the specification $\phi$. A *probabilistic specification* is then defined as an ordered pair $(\phi, p)$ of a specification $\phi$ and a probability parameter $p \in [0, 1]$ with which the specification needs to be satisfied. We say that the system satisfies the probabilistic specification $(\phi, p)$ at the initial state $\mathbf{x}_0 \in \mathcal{X}$ if the probability of any trajectory in $\Omega_{\mathbf{x}_0}$ satisfying $\phi$ is at least $p$, i.e. if $\mathbb{P}_{\mathbf{x}_0}[\rho \in \Omega_{\mathbf{x}_0} \mid \rho \models \phi] \geq p$.

**SPECTRL and abstract graphs** Reach-avoid specifications are one of the most common and practically relevant specifications appearing in safety-critical applications that generalize both reachability and safety specifications [54]. Given a target region and an unsafe region, a reach-avoid specifications requires that a system controlled by a policy reaches the target region while avoiding the unsafe region. The SPECTRL specification language [32] allows specifications of reachability and avoidance objectives as well as their sequential or disjunctive composition. See Appendix A for the formal definition of the SPECTRL syntax and semantics.[2]

It was shown in [33] that a SPECTRL specification can be translated into an abstract graph over reach-avoid specifications. Intuitively, an *abstract graph* is a directed acyclic graph (DAG) whose vertices represent regions of system states and whose edges are annotated with a safety specification. Hence, each edge can be associated with a *reach-avoid specification* where the goal is to drive the system from the region corresponding to the source vertex of the edge to the region corresponding to the target vertex of the edge while satisfying the annotated safety specification.

**Definition 1** (Abstract graph). *An* abstract graph $G = (V, E, \beta, s, t)$ *is a DAG, where $V$ is the vertex set, $E$ is the edge set, $\beta : V \cup E \to \mathcal{B}(\mathcal{X})$ maps each vertex and each edge to a* subgoal region *in $\mathcal{X}$, $s \in V$ is the source vertex and $t \in V$ is the target vertex. Furthermore, we require that $\beta(s) = \mathcal{X}_0$.*

---

[2]In Appendix A, we also show that SPECTRL strictly subsumes all specifications belonging to the Finitary fragment of LTL, that has been shown to be the PAC-MDP-learnable fragment of LTL [66]. However, the converse is not true and SPECTRL is strictly more general since avoidance is not finitary.

Given a trajectory $\rho = (\mathbf{x}_t, \mathbf{u}_t, \omega_t)_{t=0}^{\infty}$ of the system and an abstract graph $G = (V, E, \beta, s, t)$, we say that $\rho$ satisfies *abstract reachability* for $G$ (written $\rho \models G$) if it gives rise to a path in $G$ that traverses $G$ from $s$ to $t$ and satisfies reach-avoid specifications of all traversed edges. We formalize this notion in Appendix B. It was shown in [33, Theorem 3.4] that for each SPECTRL specification $\phi$ once can construct an abstract graph $G$ such that for each trajectory $\rho$ we have $\rho \models \phi$ iff $\rho \models G$. For completeness, we provide this construction in Appendix B.

**Problem Statement** We now formally define the problem that we consider in this work. Consider a stochastic feedback loop system defined as above and let $\mathcal{X}_0 \subseteq \mathcal{X}$ be the set of initial states. Let $(\phi, p)$ be a probabilistic specification with $\phi$ being a specification formula in SPECTRL and $p \in [0, 1]$ being a probability threshold. Then, our goal is to learn a policy $\pi$ such that the stochastic feedback loop system controlled by the policy $\pi$ *guarantees* satisfaction of the probabilistic specification $(\phi, p)$ at every initial state $\mathbf{x}_0 \in \mathcal{X}_0$, i.e. that for every $\mathbf{x}_0 \in \mathcal{X}_0$ we have $\mathbb{P}_{\mathbf{x}_0}[\rho \in \Omega_{\mathbf{x}_0} \mid \rho \models \phi] \geq p$.

**Compositional learning algorithm** Our goal is not only to learn policy that guarantees satisfaction of the probabilistic specification, but also to learn such a policy in a compositional manner. Given a SPECTRL specification $\phi$, a probability threshold $p \in [0, 1]$ and an abstract graph $G$ of $\phi$, we say that a policy $\pi$ is a *compositional policy* for the probabilistic specification $(\phi, p)$ if it guarantees satisfaction of $(\phi, p)$ and is obtained by composing a number of *edge policies* learned for reach-avoid tasks associated to edges of $G$. An algorithm is said to be *compositional* for a probabilistic specification $(\phi, p)$ if it learns a compositional policy for the probabilistic specification $(\phi, p)$. In this work, we present a compositional algorithm for the given probabilistic specification $(\phi, p)$.

# 4   Improved Bound for Probabilistic Reach-avoidance

In this section, we define reach-avoid supermartingales (RASMs) and derive an improved lower bound on the probability of reach-avoidance that RASMs can be used to formally certify. RASMs, together with a lower bound on the probability of reach-avoidance that they guarantee and a method for learning a control policy and an RASM, were originally proposed in [68]. The novelty in this section is that we prove that RASMs imply a *strictly stronger* bound on the probability of satisfying reach-avoidance, compared to the bound derived in [68]. We achieve this by proposing an alternative formulation of RASMs, which we call *multiplicative RASMs*. In contrast to the original definition of RASMs which imposes *additive* expected decrease condition, our formulation of RASMs imposes *multiplicative* expected decrease condition. In what follows, we define multiplicative RASMs. Then, we first show in Theorem 1 that they are equivalent to RASMs. Second, we show in Theorem 2 that by analyzing the multiplicative expected decrease, we can derive an *exponentially tighter* bound on the probability of reach-avoidance. These two results imply that we can utilize the procedure of [68] to learn policies together with multiplicative RASMs that provide strictly better formal guarantees on the probability reach-avoidance.

**Prior work – reach-avoid supermartingales** RASMs [68] are continuous functions that assign real values to system states, are required to be nonnegative and to satisfy the *additive* expected decrease condition, i.e. to strictly decrease in expected value by some additive term $\epsilon > 0$ upon every one-step evolution of the system dynamics until either the target set or the unsafe set is reached. Furthermore, the initial value of the RASM is at most 1 while the value of the RASM needs to exceed some $\lambda > 1$ in order for the system to reach the unsafe set. Thus, an RASM can intuitively be viewed as an invariant of the system which shows that the system has a tendency to *converge* either to the target or the unsafe set while also being *repulsed away* from the unsafe set, where this tendency is formalized by the expected decrease condition. To emphasize the additive expected decrease, we refer to RASMs of [68] as additive RASMs.

**Definition 2** (Additive reach-avoid supermartingales [68]). *Let $\epsilon > 0$ and $\lambda > 1$. A continuous function $V : \mathcal{X} \to \mathbb{R}$ is an $(\epsilon, \lambda)$-additive reach-avoid supermartingale $((\epsilon, \lambda)$-additive RASM) with respect to $\mathcal{X}_t$ and $\mathcal{X}_u$, if:*

1. Nonnegativity condition. $V(\mathbf{x}) \geq 0$ *for each $\mathbf{x} \in \mathcal{X}$.*
2. Initial condition. $V(\mathbf{x}) \leq 1$ *for each $\mathbf{x} \in \mathcal{X}_0$.*
3. Safety condition. $V(\mathbf{x}) \geq \lambda$ *for each $\mathbf{x} \in \mathcal{X}_u$.*
4. Additive expected decrease condition. *For each $\mathbf{x} \in \mathcal{X} \backslash \mathcal{X}_t$ at which $V(\mathbf{x}) \leq \lambda$, we have* $V(\mathbf{x}) \geq \mathbb{E}_{\omega \sim d}[V(f(\mathbf{x}, \pi(\mathbf{x}), \omega))] + \epsilon$.

Table 1: Comparison of the bound in Theorem 2 and the bound in [68] for several values of $\lambda$ when $\gamma = 0.99$. We set $\Delta = 0.1$ which bounds the step size in the 9-Rooms environment, and $L_V = 5$ which is an upper bound on the Lipschitz constant that we observe in experiments. Thus, the bound in Theorem 2 is $1 - \frac{1}{\lambda} \cdot \gamma^N = 1 - \frac{1}{\lambda} \cdot 0.99^{2\lambda}$ and in [68] is $1 - \frac{1}{\lambda}$.

| $\lambda$ | 10 | 100 | 1000 |
|---|---|---|---|
| CLAPS bound (ours) | 0.91820 | 0.99866 | 0.99999 |
| [68] | 0.9 | 0.99 | 0.999 |

**Multiplicative RASMs and equivalence** In this work we introduce multiplicative RASMs which need to decrease in expected value by at least some *multiplicative* factor $\gamma \in (0, 1)$. We also impose the *strict positivity* condition outside of the target set $\mathcal{X}_t$. Note that any $(\epsilon, \lambda)$-additive RASM satisfies this condition with the lower bound $\epsilon > 0$.

**Definition 3** (Multiplicative reach-avoid supermartingales). *Let $\gamma \in (0, 1)$, $\delta > 0$ and $\lambda > 1$. A continuous function $V : \mathcal{X} \to \mathbb{R}$ is a $(\gamma, \delta, \lambda)$-multiplicative reach-avoid supermartingale ($(\gamma, \delta, \lambda)$-multiplicative RASM) with respect to $\mathcal{X}_t$ and $\mathcal{X}_u$, if:*

1. *Nonnegativity condition. $V(\mathbf{x}) \geq 0$ for each $\mathbf{x} \in \mathcal{X}$.*
2. *Strict positivity outside $\mathcal{X}_t$ condition. $V(\mathbf{x}) \geq \delta$ for each $\mathbf{x} \in \mathcal{X} \backslash \mathcal{X}_t$.*
3. *Initial condition. $V(\mathbf{x}) \leq 1$ for each $\mathbf{x} \in \mathcal{X}_0$.*
4. *Safety condition. $V(\mathbf{x}) \geq \lambda$ for each $\mathbf{x} \in \mathcal{X}_u$.*
5. *Multiplicative expected decrease condition. For each $\mathbf{x} \in \mathcal{X} \backslash \mathcal{X}_t$ at which $V(\mathbf{x}) \leq \lambda$, we have $\gamma \cdot V(\mathbf{x}) \geq \mathbb{E}_{\omega \sim d}[V(f(\mathbf{x}, \pi(\mathbf{x}), \omega))]$.*

We show that a function $V : \mathcal{X} \to \mathbb{R}$ is an additive RASM if and only if it is a multiplicative RASM.

**Theorem 1.** *[Proof in Apendix C] The following two statements hold:*

1. *If a continuous function $V : \mathcal{X} \to \mathbb{R}$ is an $(\epsilon, \lambda)$-additive RASM, then it is also a $(\frac{\lambda - \epsilon}{\lambda}, \min\{\epsilon, \lambda\}, \lambda)$-multiplicative RASM.*
2. *If a continuous function $V : \mathcal{X} \to \mathbb{R}$ is a $(\gamma, \delta, \lambda)$-multiplicative RASM, then it is also an $((1 - \gamma) \cdot \delta, \lambda)$-additive RASM.*

**Improved bound** The following theorem shows that the existence of a multiplicative RASM implies a lower bound on the probability with which the system satisfies the reach-avoid specification.

**Theorem 2.** *[Proof in Appendix D] Let $\gamma \in (0, 1)$, $\delta > 0$ and $\lambda > 1$, and suppose that $V : \mathcal{X} \to \mathbb{R}$ is a $(\gamma, \delta, \lambda)$-multiplicative RASM with respect to $\mathcal{X}_t$ and $\mathcal{X}_u$. Suppose furthermore that $V$ is Lipschitz continuous with a Lipschitz constant $L_V$, and that the system under policy $\pi$ satisfies the* bounded step property, *i.e. that there exists $\Delta > 0$ such that $||\mathbf{x} - f(x, \pi(\mathbf{x}), \omega)||_1 \leq \Delta$ holds for each $\mathbf{x} \in \mathcal{X}$ and $\omega \in \mathcal{W}$. Let $N = \lfloor (\lambda - 1)/(L_V \cdot \Delta) \rfloor$. Then, for every $\mathbf{x}_0 \in \mathcal{X}_0$, we have that*

$$\mathbb{P}_{\mathbf{x}_0}\Big[ReachAvoid(\mathcal{X}_t, \mathcal{X}_u)\Big] \geq 1 - \frac{1}{\lambda} \cdot \gamma^N.$$

Here, $N = \lfloor (\lambda - 1)/(L_V \cdot \Delta) \rfloor$ is the smallest number of time steps in which the system could hypothetically reach the unsafe set $\mathcal{X}_u$. This is because, for the system to violate safety, by the Initial and the Safety conditions of multiplicative RASMs the value of $V$ must increase from at most 1 to at least $\lambda$. But the value of $V$ can increase by at most $L_V \cdot \Delta$ in any single time step since $\Delta$ is the maximal step size of the system and $L_V$ is the Lipschitz constant of $V$. Hence, the system cannot reach $\mathcal{X}_u$ in less than $N = \lfloor (\lambda - 1)/(L_V \cdot \Delta) \rfloor$ time steps.

We conclude this section by comparing our bound to the bound $\mathbb{P}_{\mathbf{x}_0}[ReachAvoid(\mathcal{X}_t, \mathcal{X}_u)] \geq 1 - \frac{1}{\lambda}$ of [68] for additive RASMs. Our bound on the probability of violating reach-avoidance is tighter by a factor of $\gamma^N$. Notice that this term decays exponentially as $\lambda$ increases, hence our bound is *exponentially tighter* in $\lambda$. This is particularly relevant if we want to verify reach-avoidance with high probability that is close to 1, as it allows using a much smaller value of $\lambda$ and significantly relaxing the Safety condition in Definition 3. To evaluate the quality of the bound improvement on an example, in Table 1 we consider the 9-Rooms environment and compare the bounds for multiple values of $\lambda$ when $\gamma = 0.99$. Note that $\Delta$ is the property of the system and not the RASM. Thus, the value of $\lambda$ is the key factor for controlling the quality of the bound. Results in Table 1 show that, in order to verify probability 99.9% reach-avoidance for an edge policy in the 9-Room example, our new bound allows using $\lambda \approx 100$ whereas with the old bound the algorithm needs to use $\lambda \approx 1000$.

**Remark 1** (Comparison of RASMs and stochastic barrier functions). *Stochastic barrier functions (SBFs) [45, 46] were introduced for proving probabilistic safety in stochastic dynamical systems, i.e. without the additional reachability condition as in reach-avoid specifications. If one is only interested in probabilistic safety, RASMs of [68] reduce to SBFs by letting $\mathcal{X}_t = \emptyset$ and $\epsilon = 0$ in Definition 2 for additive RASMs, and $\mathcal{X}_t = \emptyset$ and $\gamma = 1$ in Definition 3 for multiplicative RASMs. The bound on the probability of reach-avoidance $\mathbb{P}_{\mathbf{x}_0}[ReachAvoid(\emptyset, \mathcal{X}_u)] \geq 1 - \frac{1}{\lambda}$ of [68] for additive RASMs coincides with the bound on the probability of safety implied by SBFs of [45, 46]. Hence, our novel bound in Theorem 2 also provides tighter lower bound on the safety probability guarantees via SBFs.*

*Exponential stochastic barrier functions (exponential SBFs) have been considered in [49] in order to provide tighter bounds on safety probability guarantees via SBFs for finite time horizon systems, in which the length of the time horizon is fixed and known a priori. Exponential SBFs also consider a multiplicative expected decrease condition, similar to our multiplicative RASMs and provide lower bounds on safety probability which are tighter by a factor which is exponential in the length $N$ of the time horizon [49, Theorem 2]. However, as the length of the time horizon $N \to \infty$, their bound reduces to the bound of [45, 46]. In contrast, Theorem 2 shows that our multiplicative RASMs provide a tighter lower bound on safety (or more generally, reach-avoid) probability even in unbounded (i.e. indefinite) or infinite time horizon systems.*

## 5   Compositional Learning for Probabilistic Specifications

We now present the CLAPS algorithm for learning a control policy that guarantees satisfaction of a probabilistic specification $(\phi, p)$, where $\phi$ is a SPECTRL formula. The idea behind our algorithm is as follows. The algorithm first translates $\phi$ into an abstract graph $G = (V, E, \beta, s, t)$ using the translation discussed in Section 3, in order to decompose the problem into a series of reach-avoid tasks associated to abstract graph edges. The algorithm then solves these reach-avoid tasks by learning policies for abstract graph edges. Each edge policy is learned together with a (multiplicative) RASM which provides formal guarantees on the probability of satisfaction of the reach-avoid specification proved in Theorem 2. Finally, the algorithm combines these edge policies in order to obtain a global policy that guarantees satisfaction of $(\phi, p)$. The algorithm pseudocode is presented in Algorithm 1. We assume the POLICY+RASM subprocedure which, given a reach-avoid task and given a probability parameter $p'$, learns a control policy together with a RASM that proves that the policy guarantees reach-avoidance with probability at least $p'$. Due to the equivalence of additive and multiplicative RASMs that we proved in Theorem 1, for this we can use the procedure of [68] as an off-the-shelf method. For completeness of our presentation, we also provide details behind the POLICY+RASM subprocedure in Appendix E.

**Challenges** There are two important challenges in designing an algorithm based on the above idea. First, it is not immediately clear how probabilistic reach-avoid guarantees provided by edge policies may be used to deduce the probability with which the global specification $\phi$ is satisfied. Second, since SPECTRL formulas allow for disjunctive specifications, it might not be necessary to solve reach-avoid tasks associated to each abstract graph edge and naïvely doing so might lead to inefficiency. Our algorithm solves both of these challenges by first *topologically ordering* the vertices of the abstract graph and then performing a *forward pass* in which vertices are processed according to the topological ordering. The forward pass is executed in a way which allows our algorithm to keep track of the cumulative probability with which edge policies that have already been learned might violate the global specification $\phi$, therefore also allowing the algorithm to learn edge policies *on-demand* and to identify abstract graph edges for which solving reach-avoid tasks would be redundant.

**Forward pass on the abstract graph** Since an abstract graph is a directed acyclic graph (DAG), we perform a topological sort on $G$ in order to produce an ordering $s = v_1, v_2, \ldots, v_{|V|} = t$ of its vertices (line 4 in Algorithm 1). This ordering satisfies the property that each edge $(v_i, v_j)$ in $G$ must satisfy $i < j$, i.e. any each edge must be a "forward edge" with respect to topological ordering. Algorithm 1 now initializes an empty dictionary $\mathrm{Prob}$ and sets $\mathrm{Prob}[s] = 1$ for the source vertex $s$ (line 5) and performs a forward pass to process the remaining vertices in the abstract graph according to the topological ordering (lines 6-13). The purpose of the dictionary $\mathrm{Prob}$ is to store lower bounds on the probability with which previously processed abstract graph vertices are reached by following the computed edge policies while satisfying reach-avoid specifications of the traversed abstract graph edges. For each newly processed vertex $v_i$, Algorithm 1 learns policies for the reach-avoid tasks associated to all abstract graph edges that are incoming to $v_i$. It then combines probabilities with

---

**Algorithm 1** Compositional Learning for Probabilistic Specifications (CLAPS )

---

1: **Input** Dynamics function $f$, probability distribution $d$, initial set $\mathcal{X}_0$,
2:        SPECTRL specification $\phi$, probability threshold $p \in [0, 1]$
3: $G = (V, E, \beta, s, t) \leftarrow$ abstract graph for $\phi$
4: $s = v_1, v_2, \ldots, v_{|V|} = t \leftarrow$ topological ordering of vertices in $G$
5: $\text{Prob} \leftarrow$ empty hash map, $\text{Prob}[s] \leftarrow 1$
6: **for** i = 2, 3, . . . |V| **do**
7:     $\text{Prob}[v_i] \leftarrow 0$
8:     $N(v_i) \leftarrow \{v \in V \mid (v, v_i) \in E \wedge \text{Prob}[v] \geq p\}$
9:     **for all** $v \in N(v_i)$ **do**
10:       $\pi_{(v,v_i)}$, $p_{(v,v_i)} \leftarrow$ use binary search to find the maximal probability $p_{(v,v_i)}$ for which POLICY+RASM manages to learn a policy $\pi_{(v,v_i)}$ that solves the reach-avoid task associated to edge $(v, v_i)$ with probability at least $p_{(v,v_i)}$
11:       $\text{Prob}[v_i] \leftarrow \max\{\text{Prob}[v_i],\ p_{(v,v_i)} \cdot \text{Prob}[v]\}$
12:     **end for**
13: **end for**
14: **if** $\text{Prob}[t] \geq p$ **then**
15:     $\pi \leftarrow$ global policy obtained by composing edge policies
16:     **return** Policy $\pi$ that guarantees satisfaction of $(\phi, p)$.
17: **else**
18:     **return** Policy for the probabilistic specification $(\phi, p)$ could not be learned.
19: **end if**

---

which the learned edge policies satisfy associated reach-avoid tasks with the probabilities that have been stored for the previously processed vertices in order to compute the lower bound $\text{Prob}[v_i]$.

More concretely, for each $2 \leq i \leq |V|$, Algorithm 1 first stores $\text{Prob}[v_i] = 0$ as a trivial lower bound (line 7). It then computes the set $N(v_i)$ of all predecessors of $v_i$ for which previously learned edge policies ensure reachability with probability at least $p$ while satisfying reach-avoid specification of all traversed abstract graph edged (line 8). Recall, $p$ is the global probability threshold with which the agent needs to satisfy the SPECTRL specification $\phi$, and since Algorithm 1 processes vertices according to the topological ordering all vertices in $N(v_i)$ have already been processed. If $N(v_i)$ is empty, then the algorithm concludes that $v_i$ cannot be reached in the abstract graph while satisfying reach-avoid specifications of traversed edges with probability at least $p$. In this case, the algorithm does not try to learn edge policies for the edges with the target vertex $v_i$ and proceeds to processing the next vertex in the topological ordering. Thus, Algorithm 1 ensures that edge policies are learned *on-demand*. If $N(v_i)$ is not empty, then for each $v \in N(v_i)$ the algorithm uses POLICY+RASM together with binary search to find the maximal probability $p_{(v,v_i)}$ for which POLICY+RASM manages to learn a policy $\pi_{(v,v_i)}$ that solves the reach-avoid task with probability at least $p_{(v,v_i)}$ (line 10). We run the binary search only up to a pre-specified precision, which is a hyperparameter of our algorithm. Since the reach-avoid specification of the edge $(v, v_i)$ is satisfied with probability at least $p_{(v,v_i)}$ and since $v$ is reached in the abstract graph while satisfying reach-avoid specification of all traversed edges with probability at least $\text{Prob}[v]$, Algorithm 1 updates its current lower bound $\text{Prob}[v_i]$ for $v_i$ to $p_{(v,v_i)} \cdot \text{Prob}[v]$ whenever $p_{(v,v_i)} \cdot \text{Prob}[v]$ exceeds the stored bound $\text{Prob}[v_i]$ (line 11). This procedure is repeated for each $v \in N(v_i)$.

**Composing edge policies into a global policy** Once all abstract graph vertices have been processed, Algorithm 1 checks if $\text{Prob}[t] \geq p$ (line 14). If so, it composes the learned edge policies into a global policy $\pi$ that guarantees satisfaction of the probabilistic specification $(\phi, p)$ (line 15) and returns the policy $\pi$ (line 16). The composition is performed as follows. First, note that for each vertex $v \neq s$ in the abstract graph for which $\text{Prob}[v] > 0$, by lines 7-11 in Algorithm 1 there must exist a vertex $v' \in N(v)$ such that the algorithm has successfully learned a policy $\pi_{(v',v)}$ for the edge $(v', v)$ and such that $\text{Prob}[v] = p_{(v',v)} \cdot \text{Prob}[v']$. Hence, repeatedly picking such predecessors yields a finite path $s = v_{i_0}, v_{i_1}, \ldots, v_{i_k} = t$ in the abstract graph such that, for each $1 \leq j \leq k$, the algorithm has successfully learned an edge policy $\pi_{(v_{i_{j-1}}, v_{i_j})}$ and such that $\text{Prob}[v_{i_j}] = p_{(v_{i_{j-1}}, v_{i_j})} \cdot \text{Prob}[v_{i_{j-1}}]$ holds. The global policy $\pi$ starts by following the edge policy $\pi_{(v_{i_0}, v_{i_1})}$ until the target region $\beta(v_{i_1})$ is reached or until the safety constraint $\beta(v_{i_0}, v_{i_1})$ associated to the edge is violated. Then,

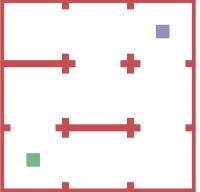

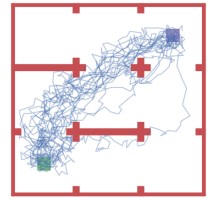

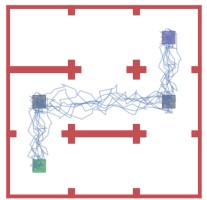

(a) The objective is to navigate from the initial set in the bottom left (in green) to the target set in the top right room (in violet).

(b) Visualization of trajectories of an end-to-end learned policy. The policy hits the walls and cannot be verified as safe.

(c) Visualization of trajectories of the three verifiable compositional policies with intermediate initial/target states shown in grey.

Figure 1: Stochastic Nine Rooms environment for which an end-to-end learned policy cannot be verified safe while our algorithm is able to decompose the task into three verifiable subtasks.

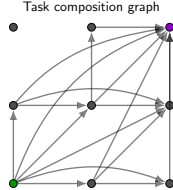
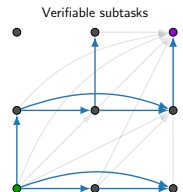
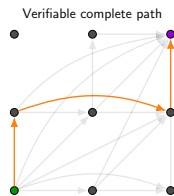

Figure 2: Decomposition of the Stochastic Nine Rooms task into subtasks. Left: Individual subtasks shown as graph edges. Center: Successfully verifiable subtasks are shown in blue edges, whereas the verification procedure of the gray edges failed. Right: Verified path for solving the complete task.

it follows the edge policy $\pi_{(v_{i_1}, v_{i_2})}$ until the target region $\beta(v_{i_2})$ is reached or until the safety constraint $\beta(v_{i_1}, v_{i_12})$ associated to the edge is violated. This is repeated for each edge along the path $s = v_{i_0}, v_{i_1}, \ldots, v_{i_k} = t$ until the system reaches a state within the target region $\beta(t)$. If at any point the safety constraint is violated prior to reaching the target region associated to the edge, policy $\pi$ follows the current edge policy indefinitely and no longer updates it as the specification $\phi$ has been violated. As we prove in Theorem 3, violation of the safety constraint associated to some edge or the system never reaching the target region associated to some edge happen with probability at most $1 - p$. If $\mathrm{Prob}[t] < p$, then Algorithm 1 returns that it could not learn a policy that guarantees satisfaction of $(\phi, p)$. Theorem 3 establishes the correctness of Algorithm 1 and its proof shows that the global policy $\pi$ obtained as above indeed satisfies the probabilistic specification $(\phi, p)$.

**Theorem 3.** *[Proof in Appendix F] Algorithm 1 is compositional, and if it outputs a policy $\pi$, then $\pi$ guarantees the probabilistic specification $(\phi, p)$.*

## 6 Experiments

We implement a prototype of Algorithm 1 to validate its effectiveness [3]. Motivated by the experiments of [33], we define an environment in which an agent must navigate safely between different rooms. However, different from [33] our Stochastic Nine Rooms environment perturbs the agent by random noise in each step. Concretely, the system is governed by the dynamics function, $x_{t+1} = x_t + 0.1 \min(\max((a_t, -1), 1) + \omega_t$, where $\omega_t$ is drawn from a triangular noise distribution. The state space of the environment is defined as $[0, 3] \times [0, 3] \subset \mathbb{R}^2$. The set of the initial states is $[0.4, 0.6] \times [0.4, 0.6]$, whereas the target states are defined as $[2.4, 2.6] \times [2.4, 2.6]$. The state space contains unsafe areas that should be avoided, i.e., the red *walls* shown in Figure 1. Specifically, these unsafe areas cannot be entered by the agent, and coming close to them during the RL training incurs a penalty. Note that this defines a reach-avoid task where the goal is to reach the target region from the initial region while avoiding the unsafe regions defined by red walls.

We first run the POLICY+RASM procedure (i.e. the method of [68]) on this task directly and observe that it is not able to learn a policy with any formal guarantees on probability of reach-avoidance.

---

[3]Our code is available at `https://github.com/mlech26l/neural_martingales`

We then translate this task into a SPECTRL task which decomposes it into a set of simpler reach-avoid tasks by constructing an abstract graph as follows. The vertex set of the abstract graph consists of centers of each of the 9 *rooms*, and the target regions associated to each room center are $[0.4 + x, 0.6 + x] \times [0.4 + y, 0.6 + y]$ for $x, y \in \{0, 1, 2\}$. Edges of this abstract graph are shown in Figure 2 left. To each edge, we associate an unsafe region to be avoided by taking red walls incident to rooms in which the edge is contained.

For each reach-avoid edge task, we run the proximal policy optimization (PPO) [50] reinforcement learning algorithm to initialize policy parameters. We then run the POLICY+RASM procedure to learn a policy and a RASM which proves probabilistic reach-avoidance. We set the timeout to 4 hours for each reach-avoid subtask. In Figure 1 b), we visualize the PPO policy that was trained to reach the target states directly from the initial states. As shown, the policy occasionally hits a wall and cannot be verified, i.e., POLICY+RASM procedure times out. In Figure 2 center, we visualize all successfully verified subtasks in blue. The reach-avoid lower bounds for each of the subtasks are listed in Table 2. Note that the simplest decomposition passing over the bottom right room could not be verified, which shows that the systematic decomposition used in our algorithm has advantages over manual task decompositions. Finally, in Figure 2 right, we highlight the path from the initial states to the target states via safe intermediate sets. In Figure 1 c), we visualize the trajectories of these three compositional policies. As shown, the trajectories never reach an unsafe area.

Table 2: Verified lower bounds on reach-avoid probability for each edge policy. Initial and target states are represented by pairs $(x, y)$ with $[0.4 + x, 0.6 + x] \times [0.4 + y, 0.6 + y]$ for $x, y \in \{0, 1, 2\}$.

| Start $(x_1, y_1)$ | Goal $(x_2, y_2)$ | Reach-avoidance probability |
|---|---|---|
| (0,0) | (1,0) | 75.1% |
| (0,0) | (2,0) | 57.7% |
| (0,0) | (0,1) | 72.3% |
| (1,0) | (2,0) | 71.4% |
| (0,1) | (2,1) | 61.5% |
| (0,1) | (1,1) | 72.2% |
| (1,1) | (1,2) | 73.3% |
| (1,2) | (2,1) | 68.5% |
| (2,1) | (2,2) | 74.5% |
| (0,0) | (2,2) | Fail (method of [68]) |
| (0,0) | (2,2) | 33.0% (our method) |

# 7 Concluding Remarks

We propose CLAPS, a novel method for learning and verifying a composition of neural network policies for stochastic control systems. Our method considers control tasks under specifications in the SPECTRL lanugange, decomposes the task into an abstract graph of reach-avoid tasks and uses reach-avoid supermartingales to provide formal guarantees on the probability of reach-avoidance in each subtask. We also prove that RASMs imply a strictly tighter lower bound on the probability of reach-avoidance compared to prior work. These results provide confidence in the ability of CLAPS to find meaningful guarantees for global compositional policies, as shown by experimental evaluation in the Stochastic Nine Rooms environment. While our approach has significant conceptual, theoretical and algorithmic novelty, it is only applicable to SPECTRL logical specifications and not to the whole of LTL. Furthermore, our algorithm is not guaranteed to return a policy and does not guarantee tight bounds on the probability of satisfying the logical specification. Overcoming these limitations will provide further confidence in deploying RL based solutions for varied applications.

## Acknowledgments

This work was supported in part by the ERC-2020-AdG 101020093 (VAMOS) and the ERC-2020-CoG 863818 (FoRM-SMArt).

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

# Appendix

## A   SPECTRL syntax and semantics and the proof of Theorem 4

**Syntax** A specification in SPECTRL is defined in terms of *predicates* and *specification formulas*. An atomic predicate is a boolean function $a : \mathcal{X} \to \{\text{true}, \text{false}\}$ which for each system state specifies whether it satisfies the predicate, and a predicate is defined as a boolean combination of atomic predicates. Specification formulas in SPECTRL are then defined by the grammar

$$\phi := \text{achieve } b \mid \phi_1 \text{ ensuring } b \mid \phi_1; \phi_2 \mid \phi_1 \text{ or } \phi_2 \tag{1}$$

where $b$ is a predicate and $\phi_1$ and $\phi_2$ are specification formulas. Intuitively, achieve $b$ requires the agent to reach a state in which the predicate $b$ is satisfied and $\phi_1$ ensuring $b$ requires the agent to satisfy the specification $\phi$ while only visiting states in which the predicate $b$ is satisfied. The clause $\phi_1; \phi_2$ requires the agent to first satisfy specification $\phi_1$ and then satisfy specification $\phi_2$. Finally, $\phi_1$ or $\phi_2$ requires satisfaction of at least one of the specifications $\phi_1$ or $\phi_2$.

**Semantics** Given a trajectory $\rho = (\mathbf{x}_t, \mathbf{u}_t, \omega_t)_{t=0}^{\infty}$ and writing $\rho^K = (\mathbf{x}_t, \mathbf{u}_t, \omega_t)_{t=0}^{K}$ for its finite prefix of length $K$, the semantics of each SPECTRL clause are formally defined as follows:

$$
\begin{aligned}
\rho &\models \phi & &\exists K \in \mathbb{N}_0 \text{ s.t. } \rho^K \models \phi \\
\rho^K &\models \text{achieve } p & &\exists t \leq K \text{ s.t. } p(\rho_t^K) = \text{true} \\
\rho^K &\models \phi_1 \text{ ensuring } p & &\rho^K \models \phi_1 \wedge \forall t \leq K. \, p(\rho_t^K) = \text{true} \\
\rho^K &\models \phi_1; \phi_2 & &\exists t \leq K \text{ s.t. } \rho_{[0:t]}^K \models \phi_1 \text{ and } \rho_{[t:K]}^K \models \phi_2 \\
\rho^K &\models \phi_1 \text{ or } \phi_2 & &\rho^K \models \phi_1 \text{ or } \rho^K \models \phi_2
\end{aligned}
$$

Here, $\rho_t^K$ denotes the $t$-th state along $\rho^K$, $\rho_{[0:t]}^K$ denotes the prefix of $\rho^K$ consisting of the first $t + 1$ states along $\rho^K$ and $\rho_{[t:K]}^K$ denotes the suffix of $\rho^K$ that starts in the $(t + 1)$-st state along $\rho^K$.

**Theorem 4.** *For each $\phi \in$ Finitary there exists $\phi' \in$ SPECTRL such that, for any word $w$, the word $w$ is accepted by $\phi$ iff the word $w$ is accepted by $\phi'$.*

*Proof.* Let $\phi$ be a finitary specification defined over the set of atomic predicates $AP$. Since $\phi$ is finitary, there exist a finite time horizon $H$ and a set $L$ of words over $AP$ of length $H$ such that an infinite word over the alphabet $AP$ is accepted by $\phi$ iff its prefix of length $H$ is contained in $L$. Define a SPECTRL formula $\phi'$ via:

$$\phi' = \bigvee_{(w_1, \dots, w_H) \in L} p(w_1); \, p(w_2); \, \dots; \, p(w_H)$$

where each $p(w_i)$ is an atomic predicate associated to the $i$-th letter in the word $(w_1, \dots, w_H)$ and ; denotes sequential composition of SPECTRL specifications. Then, an infinite word $w$ is accepted by $\phi$ if and only if the prefix of $w$ of length $H$ is contained in $L$, which holds if and only if $w$ is accepted by the SpectRL formula $\phi'$. This completes our reduction. $\square$

## B   Abstract Reachability Definition and Proof of Theorem 5

Given a trajectory $\rho = (\mathbf{x}_t, \mathbf{u}_t, \omega_t)_{t=0}^{\infty}$ of the system and an abstract graph $G = (V, E, \beta, s, t)$, we say that $\rho$ satisfies *abstract reachability* for $G$ (written $\rho \models G$) if it gives rise to a path in $G$ that traverses $G$ from $s$ to $t$ and satisfies reach-avoid specifications of all traversed edges. Formally, we require that there exists a sequence of time steps $0 = i_0 < i_1 < \dots < i_k$ and a finite path $s = v_0, v_1, \dots, v_k = t$ in $G$ such that

1. $\mathbf{x}_{i_j} \in \beta(v_j)$ holds for each $0 \leq j \leq k$, and
2. $\mathbf{x}_t \in \beta(v_j, v_{j+1})$ holds for each $0 \leq j < k$ and $i_j \leq t \leq i_{j+1}$.

Intuitively, the first condition encodes that the trajectory satisfies reachability specifications of traversed vertices in $G$ while the second condition encodes that it satisfies avoidance specifications of traversed edges in $G$. We then say that a policy $\pi$ for the system satisfies *abstract reachability for $G$ with probability* $p \in [0, 1]$ at an initial state $\mathbf{x}_0 \in \mathcal{X}_0$, if we have that $\mathbb{P}_{\mathbf{x}_0}[\rho \in \Omega_{\mathbf{x}_0} \mid \rho \models G] \geq p$.

We now provide the proof of Theorem 5.

**Theorem 5.** *Consider a stochastic feedback loop system with an initial set of states $\mathcal{X}_0 \subseteq \mathcal{X}$ and let $\phi$ be a SPECTRL specification. Then there exists an abstract graph $G = (V, E, \beta, s, t)$ with $|V|$ in $\mathcal{O}(|\phi|)$ such that, for each trajectory $\rho$ of the system, we have $\rho \models \phi$ if and only if $\rho \models G$. Hence, for each policy $\pi$ and initial state $\mathbf{x}_0 \in \mathcal{X}_0$, we have $\mathbb{P}_{\mathbf{x}_0}[\rho \in \Omega_{\mathbf{x}_0} \mid \rho \models \phi] = \mathbb{P}_{\mathbf{x}_0}[\rho \in \Omega_{\mathbf{x}_0} \mid \rho \models G]$.*

*Proof.* Given a SPECTRL specification $\phi$, one can construct an abstract graph $G$ such that for each trajectory $\rho$ of the system we have $\rho \models \phi$ iff $\rho \models G$ as follows. First, the specification $\phi$ is parsed according to the grammar of SPECTRL in eq. 1 in order to construct the parse tree of $\phi$. We then start by constructing an abstract graph for each leaf formula in the parse tree, and traverse the parse tree bottom-up in order to construct abstract graphs of parent formulas. The abstract graph of the specification $\phi$ is then obtained by taking the abstract graph constructed for the root in the parse tree. The leaves of the parse tree are formulas of the form achieve $p$, for which we construct an abstract graph with two vertices $s$ and $t$, a single edge $e = (s, t)$ and set $\beta(s) = \mathcal{X}_0$, $\beta(t) = \{\mathbf{x} \in \mathcal{X} \mid p(\mathbf{x}) = \text{true}\}$ and $\beta(e) = \mathcal{X}$. For a formula $\phi_1$ ensuring $p$, we take an abstract graph $(V, E, \beta, s, t)$ for the specification $\phi_1$ which was already constructed for the child node and define the abstract graph $G = (V, E, \beta', s, t)$ by simply modifying the map $\beta$ via $\beta'(e) = \beta(e) \cap \{\mathbf{x} \in \mathcal{X} \mid p(\mathbf{x}) = \text{true}\}$ for each $e \in E$. For a formula $\phi_1; \phi_2$, we take the abstract graph of the specifications $\phi_1$ and $\phi_2$ which were already constructed for the child nodes and merge them by identifying the target node of $\phi_1$ with the source node of $\phi_2$ and using the region associated to it by the abstract graph of $\phi_2$. Finally, for a formula $\phi_1$ or $\phi_2$, we introduce a novel source node $s$ with $\beta(s) = \mathcal{X}_0$, take the abstract graph of $\phi_1$ and $\phi_2$ and connect the novel source node $s$ to them by an edge. Note that this construction yields a graph with $V$ in $\mathcal{O}(|\phi|)$.

Since the above construction soundly encodes the semantics of each SPECTRL grammar element as a reach-avoid specification, it follows by induction on the depth of the parse tree that for each trajectory $\rho$ of the system we have $\rho \models \phi$ iff $\rho \models G$. The claim of Theorem 5 follows. $\square$

## C Proof of Theorem 1

**Theorem 1.** *[Proof in Apendix C] The following two statements hold:*

1. *If a continuous function $V : \mathcal{X} \to \mathbb{R}$ is an $(\epsilon, \lambda)$-additive RASM, then it is also a $(\frac{\lambda - \epsilon}{\lambda}, \min\{\epsilon, \lambda\}, \lambda)$-multiplicative RASM.*
2. *If a continuous function $V : \mathcal{X} \to \mathbb{R}$ is a $(\gamma, \delta, \lambda)$-multiplicative RASM, then it is also an $((1 - \gamma) \cdot \delta, \lambda)$-additive RASM.*

*Proof.*

1. Let $\delta = \min\{\epsilon, \lambda\}$ and $\gamma = \frac{\lambda - \epsilon}{\lambda}$. To show that $V$ is a $(\gamma, \delta, \lambda)$-multiplicative RASM, we need to show that the Strict positivity outside $\mathcal{X}_t$ and the Multiplicative expected decrease conditions hold. By the Additive expected decrease condition, for each $\mathbf{x} \in \mathcal{X} \backslash \mathcal{X}_t$ at which $V(\mathbf{x}) \leq \lambda$ we have $V(\mathbf{x}) \geq \epsilon$. So as $\delta = \min\{\epsilon, \lambda\}$, the Strict positivity outside $\mathcal{X}_t$ follows. On the other hand, observe that for every $\mathbf{x} \in \mathcal{X} \backslash \mathcal{X}_t$ at which $V(\mathbf{x}) \leq \lambda$, we have

$$\frac{\mathbb{E}_{\omega \sim d}[V(f(\mathbf{x}, \pi(\mathbf{x}), \omega))]}{V(\mathbf{x})} \leq \frac{V(\mathbf{x}) - \epsilon}{V(\mathbf{x})} \leq \frac{\lambda - \epsilon}{\lambda} = \gamma,$$

   where the first inequality follows by the Additive expected decrease condition and the second inequality follows since $\frac{z - \epsilon}{z}$ is monotonically increasing on the domain $z > \epsilon$. Hence, the Multiplicative expected decrease condition holds.

2. Let $\epsilon = (1 - \gamma) \cdot \delta$. To show that $V$ is an $(\epsilon, \lambda)$-additive RASM, we need to show that the Additive expected decrease condition holds. We show this by observing that, for each $\mathbf{x} \in \mathcal{X} \backslash \mathcal{X}_t$ such that $V(\mathbf{x}) \leq \lambda$, we have

$$V(\mathbf{x}) - \mathbb{E}_{\omega \sim d}[V(f(\mathbf{x}, \pi(\mathbf{x}), \omega))] \geq V(\mathbf{x}) - \gamma \cdot V(\mathbf{x}) = (1 - \gamma) \cdot V(\mathbf{x}) \geq (1 - \gamma) \cdot \delta,$$

   where the first inequality holds by the Multiplicative expected decrease condition and the last inequality holds by the Strict positivity outside $\mathcal{X}_t$ condition. Hence, the Additive expected decrease condition is satisfied. $\square$

# D    Proof of Theorem 2

We first provide an overview of definitions and results from martingale theory that we use in the proof. We then present the proof.

**Probability theory**    A *probability space* is a triple $(\Omega, \mathcal{F}, \mathbb{P})$ of a state space $\Omega$, a sigma-algebra $\mathcal{F}$ and a probability measure $\mathbb{P}$ that satisfies Kolmogorov axioms [64]. A *random variable* in $(\Omega, \mathcal{F}, \mathbb{P})$ is a function $X : \Omega \to \mathbb{R}$ that is $\mathcal{F}$-measurable, i.e. for each $a \in \mathbb{R}$ we have $\{\omega \in \Omega \mid X(\omega) \leq a\} \in \mathcal{F}$. A *(discrete-time) stochastic process* is a sequence $(X_i)_{i=0}^{\infty}$ of random variables in $(\Omega, \mathcal{F}, \mathbb{P})$.

**Conditional expectation**    Let $X$ be a random variable in $(\Omega, \mathcal{F}, \mathbb{P})$. Given a sub-sigma-algebra $\mathcal{F}' \subseteq \mathcal{F}$, a *conditional expectation* of $X$ given $\mathcal{F}'$ is an $\mathcal{F}'$-measurable random variable $Y$ such that, for each $A \in \mathcal{F}'$, we have

$$\mathbb{E}[X \cdot \mathbb{I}(A)] = \mathbb{E}[Y \cdot \mathbb{I}(A)].$$

Here, $\mathbb{I}(A) : \Omega \to \{0, 1\}$ is an *indicator function* of $A$, given by $\mathbb{I}(A)(\omega) = 1$ if $\omega \in A$, and $\mathbb{I}(A)(\omega) = 0$ if $\omega \notin A$. Intuitively, conditional expectation of $X$ given $\mathcal{F}'$ is an $\mathcal{F}'$-measurable random variable that behaves like $X$ upon evaluating its expected value on events in $\mathcal{F}'$. It is known that every nonnegative random variable admits a conditional expectation [64]. Moreover, the conditional expectation is almost-surely unique, meaning that for any two $\mathcal{F}'$-measurable random variables $Y$ and $Y'$ which are conditional expectations of $X$ given $\mathcal{F}'$ we have $\mathbb{P}[Y = Y'] = 1$. Therefore, we pick any such random variable as a canonical conditional expectation and denote it by $\mathbb{E}[X \mid \mathcal{F}']$.

**Supermartingales**    Let $(\Omega, \mathcal{F}, \mathbb{P})$ be a probability space and $(\mathcal{F}_i)_{i=0}^{\infty}$ be an increasing sequence of sub-sigma-algebras in $\mathcal{F}$, i.e. $\mathcal{F}_0 \subseteq \mathcal{F}_1 \subseteq \cdots \subseteq \mathcal{F}$. A nonnegative *supermartingale* with respect to $(\mathcal{F}_i)_{i=0}^{\infty}$ is a stochastic process $(X_i)_{i=0}^{\infty}$ such that each $X_i$ is $\mathcal{F}_i$-measurable, and $X_i(\omega) \geq 0$ and $\mathbb{E}[X_{i+1} \mid \mathcal{F}_i](\omega) \leq X_i(\omega)$ hold for each $\omega \in \Omega$ and $i \geq 0$. Intuitively, the second condition is the expected decrease condition, and it is formally captured via conditional expectation.

We now present two results from martingale theory that will be used in the proof. Let $(\Omega, \mathcal{F}, \mathbb{P})$ be a probability space and $(\mathcal{F}_i)_{i=0}^{\infty}$ be an increasing sequence of sub-$\sigma$-algebras in $\mathcal{F}$.

**Theorem 6** (Supermartingale convergence theorem [64]). *Let $(X_i)_{i=0}^{\infty}$ be a nonnegative supermartingale with respect to $(\mathcal{F}_i)_{i=0}^{\infty}$. Then, there exists a random variable $X_\infty$ in $(\Omega, \mathcal{F}, \mathbb{P})$ to which the supermartingale converges to with probability 1, i.e. $\mathbb{P}[\lim_{i \to \infty} X_i = X_\infty] = 1$.*

**Theorem 7** ( [35]). *Let $(X_i)_{i=0}^{\infty}$ be a nonnegative supermartingale with respect to $(\mathcal{F}_i)_{i=0}^{\infty}$. Then, for every $\lambda > 0$, we have*

$$\mathbb{P}\left[\sup_{i \geq 0} X_i \geq \lambda\right] \leq \frac{\mathbb{E}[X_0]}{\lambda}.$$

**Theorem 2.** *[Proof in Appendix D] Let $\gamma \in (0, 1)$, $\delta > 0$ and $\lambda > 1$, and suppose that $V : \mathcal{X} \to \mathbb{R}$ is a $(\gamma, \delta, \lambda)$-multiplicative RASM with respect to $\mathcal{X}_t$ and $\mathcal{X}_u$. Suppose furthermore that $V$ is Lipschitz continuous with a Lipschitz constant $L_V$, and that the system under policy $\pi$ satisfies the* bounded step property, *i.e. that there exists $\Delta > 0$ such that $||\mathbf{x} - f(x, \pi(\mathbf{x}), \omega)||_1 \leq \Delta$ holds for each $\mathbf{x} \in \mathcal{X}$ and $\omega \in \mathcal{W}$. Let $N = \lfloor (\lambda - 1)/(L_V \cdot \Delta) \rfloor$. Then, for every $\mathbf{x}_0 \in \mathcal{X}_0$, we have that*

$$\mathbb{P}_{\mathbf{x}_0}\left[\text{ReachAvoid}(\mathcal{X}_t, \mathcal{X}_u)\right] \geq 1 - \frac{1}{\lambda} \cdot \gamma^N.$$

*Proof.* Fix an initial state $\mathbf{x}_0 \in \mathcal{X}_0$. We need to show that $\mathbb{P}_{\mathbf{x}_0}[\text{ReachAvoid}(\mathcal{X}_t, \mathcal{X}_u)] \geq 1 - \frac{1}{\lambda} \cdot \gamma^N$ with $N = \lfloor (\lambda - 1)/(L_V \cdot \Delta) \rfloor$.

Before proceeding with the proof, we define some notions. Consider the probability space $(\Omega_{\mathbf{x}_0}, \mathcal{F}_{\mathbf{x}_0}, \mathbb{P}_{\mathbf{x}_0})$ over the set of all system trajectories that start in $\mathbf{x}_0 \in \mathcal{X}_0$. For each time step $t \in \mathbb{N}_0$, define $\mathcal{F}_{\mathbf{x}_0, t} \subseteq \mathcal{F}_{\mathbf{x}_0}$ to be a sub-$\sigma$-algebra which contains events that are defined in terms of the first $t$ states along a trajectory. Formally, for each $j \in \mathbb{N}_0$, let $C_j : \Omega_{\mathbf{x}_0} \to \mathcal{X}$ assign to each trajectory $\rho = (\mathbf{x}_t, \mathbf{u}_t, \omega_t)_{t \in \mathbb{N}_0} \in \Omega_{\mathbf{x}_0}$ the $j$-th state $\mathbf{x}_j$ along the trajectory. $\mathcal{F}_i$ is then defined as the smallest $\sigma$-algebra over $\Omega_{\mathbf{x}_0}$ with respect to which $C_0, C_1, \ldots, C_i$ are all measurable. The sequence $(\mathcal{F}_{\mathbf{x}_0, t})_{t=0}^{\infty}$ defines a filtration in $(\Omega_{\mathbf{x}_0}, \mathcal{F}_{\mathbf{x}_0}, \mathbb{P}_{\mathbf{x}_0})$.

Proceeding with the proof, we show that $V$ induces a supermartingale in the probability space $(\Omega_{\mathbf{x}_0}, \mathcal{F}_{\mathbf{x}_0}, \mathbb{P}_{\mathbf{x}_0})$ over the set of all system trajectories that start in $\mathbf{x}_0 \in \mathcal{X}_0$. For each $t \in \mathbb{N}_0$, define a random variable $X_t$ in $(\Omega_{\mathbf{x}_0}, \mathcal{F}_{\mathbf{x}_0}, \mathbb{P}_{\mathbf{x}_0})$ via

$$X_t(\rho) = \begin{cases} V(\mathbf{x}_t), & \text{if } \mathbf{x}_i \notin \mathcal{X}_t \text{ and } V(\mathbf{x}_i) < \lambda \text{ for all } 0 \leq i \leq t \\ 0, & \text{if } \mathbf{x}_i \in \mathcal{X}_t \text{ for some } 0 \leq i \leq t \text{ and } V(\mathbf{x}_j) < \lambda \text{ for all } 0 \leq j \leq i \\ \lambda, & \text{otherwise} \end{cases}$$

for each trajectory $\rho = (\mathbf{x}_t, \mathbf{u}_t, \omega_t)_{t \in \mathbb{N}_0} \in \Omega_{\mathbf{x}_0}$. Intuitively, $X_t$ is equal to $V$ at $\mathbf{x}_t$ until either the target set $\mathcal{X}_t$ is reached upon which $X_t$ is set to 0, or some $V(\mathbf{x}_t) \geq \lambda$ is reached upon which $X_t$ is set to $\lambda$. We claim that $(X_t)_{t=0}^{\infty}$ is a nonnegative supermartingale with respect to the filtration $(\mathcal{F}_{\mathbf{x}_0,t})_{t=0}^{\infty}$ in the probability space $(\Omega_{\mathbf{x}_0}, \mathcal{F}_{\mathbf{x}_0}, \mathbb{P}_{\mathbf{x}_0})$ and that it with probability 1 converges to either 0 or to a value that is $\geq \lambda$.

Clearly, each $X_t$ is nonnegative. To prove that $(X_t)_{t=0}^{\infty}$ is a supermartingale, note first that each $X_t$ is $\mathcal{F}_{\mathbf{x}_0,t}$ measurable as it is defined in terms of the ferst $t$ states along a trajectory. To show that the expected decrease condition is satisfied, we show that $\mathbb{E}_{\mathbf{x}_0}[X_{t+1} \mid \mathcal{F}_{\mathbf{x}_0,t}](\rho) \leq X_t(\rho)$ holds for each $t \in \mathbb{N}_0$ and $\rho = (\mathbf{x}_t, \mathbf{u}_t, \omega_t)_{t \in \mathbb{N}_0}$. We consider 3 cases based on the definition of $V$:

1. If $\mathbf{x}_0, \mathbf{x}_1, \ldots, \mathbf{x}_t \notin \mathcal{X}_t$ and $V(\mathbf{x}_i) < \lambda$ for each $0 \leq i \leq t$, then

   $$\mathbb{E}_{\mathbf{x}_0}[X_{t+1} \mid \mathcal{F}_{\mathbf{x}_0,t}](\rho)$$
   $$= \mathbb{E}_{\mathbf{x}_0}\left[X_{t+1} \cdot \left(\mathbb{I}(\mathbf{x}_{t+1} \notin \mathcal{X}_t \wedge V(\mathbf{x}_{t+1}) < \lambda) + \mathbb{I}(\mathbf{x}_{t+1} \in \mathcal{X}_t) + \mathbb{I}(V(\mathbf{x}_{t+1}) \geq \lambda)\right) \mid \mathcal{F}_{\mathbf{x}_0,t}\right](\rho)$$
   $$= \mathbb{E}_{\mathbf{x}_0}[X_{t+1} \cdot \mathbb{I}(\mathbf{x}_{t+1} \notin \mathcal{X}_t) \mid \mathcal{F}_{\mathbf{x}_0,t}](\rho) + 0 + \lambda \cdot \mathbb{E}[\mathbb{I}(V(\mathbf{x}_{t+1}) \geq \lambda) \mid \mathcal{F}_{\mathbf{x}_0,t}](\rho)$$
   $$\leq \mathbb{E}_{\omega \sim d}[V(f(\mathbf{x}_t, \mathbf{u}_t, \omega_t)) \cdot \mathbb{I}(\mathbf{x}_{t+1} \notin \mathcal{X}_t \wedge V(\mathbf{x}_{t+1}) < \lambda)]$$
   $$+ \mathbb{E}_{\omega \sim d}[V(f(\mathbf{x}_t, \mathbf{u}_t, \omega_t)) \cdot \mathbb{I}(\mathbf{x}_{t+1} \in \mathcal{X}_t)]$$
   $$+ \mathbb{E}_{\omega \sim d}[V(f(\mathbf{x}_t, \mathbf{u}_t, \omega_t)) \cdot \mathbb{I}(V(\mathbf{x}_{t+1}) \geq \lambda)]$$
   $$= \mathbb{E}_{\omega \sim d}[V(f(\mathbf{x}_t, \mathbf{u}_t, \omega_t))]$$
   $$\leq \gamma \cdot V(\mathbf{x}_t) \leq V(\mathbf{x}_t).$$

   The first equality follows by the law of total probability, the second equality follows by definition of $X_t$, the third inequality follows by observing that $V(\mathbf{x}_{t+1}) \geq X_{t+1}(\rho)$ if $\mathbf{x}_{t+1} \in \mathcal{X}_t$, the fourth equality is just the sum of expectations over disjoint sets, and finally the fifth inequality follows by the Multiplicative expected decrease condition of $V$ and the assumption that $\mathbf{x}_t \notin \mathcal{X}_t$ and $V(\mathbf{x}_t) < \lambda$.

2. If $\mathbf{x}_i \in \mathcal{X}_t$ for some $0 \leq i \leq t$ and $V(\mathbf{x}_j) < \lambda$ for all $0 \leq j \leq i$, then we have $\mathbb{E}_{\mathbf{x}_0}[X_{t+1} \mid \mathcal{F}_{\mathbf{x}_0,t}](\rho) = \gamma \cdot X_{t+1} = X_{t+1}(\rho) = 0$.

3. Otherwise, we must have $V(\mathbf{x}_i) \geq \lambda$ and $\mathbf{x}_0, \ldots, \mathbf{x}_i \notin \mathcal{X}_t$ for some $0 \leq i \leq t$, thus $\mathbb{E}_{\mathbf{x}_0}[X_{t+1} \mid \mathcal{F}_{\mathbf{x}_0,t}](\rho) = X_{t+1}(\rho) = \lambda$.

Hence, we have proved that $(X_t)_{t=0}^{\infty}$ is a nonnegative supermartingale.

By Supermartingale Convergence Theorem, we then know that $(X_t)_{t=0}^{\infty}$ with probability 1 converges to some value. We claim furthermore that this value is either 0 or $\geq \lambda$ and that the value is attained. To see this, recall that in Theorem 1 we showed that $V$ is also an $(\epsilon, \lambda)$-additive RASM with $\epsilon = (1 - \gamma) \cdot \delta$. Then, the same sequence of inequalities as in the case one above shows that $\mathbb{E}_{\mathbf{x}_0}[X_{t+1} \mid \mathcal{F}_{\mathbf{x}_0,t}](\rho) \leq X_t(\rho) - \epsilon$ if $\mathbf{x}_0, \mathbf{x}_1, \ldots, \mathbf{x}_t \notin \mathcal{X}_t$ and $V(\mathbf{x}_i) < \lambda$ for each $0 \leq i \leq t$. Thus, the limit to which $(X_t)_{t=0}^{\infty}$ converges cannot be in the open interval $(0, \lambda)$ and the claim follows.

To prove the theorem claim, we show that $(X_t)_{t=0}^{\infty}$ converges to a value with $\geq \lambda$ with probability at most $\frac{1}{\lambda} \cdot \gamma^N$. Then, since $(X_t)_{t=0}^{\infty}$ converging to 0 implies that system reaches a state in which $V < \delta$ while never reaching a state in which $V \geq \lambda$, which by the Safety and the Strict positivity outside $\mathcal{X}_t$ conditions implies that reach-avoidance is satisfied, this will imply the theorem claim. The proof until this point is analogous to the proof of [68, Theorem 1].

The technical novelty of our proof begins in the following step. We define another stochastic process $(Y_t)_{t=0}^{\infty}$ from $(X_t)_{t=0}^{\infty}$ by letting

$$Y_t = \begin{cases} X_t/\gamma^t, & \text{if } V(\mathbf{x}_i) < \lambda \text{ for all } 0 \leq i \leq t \\ Y_{t-1}, & \text{otherwise} \end{cases}$$

We claim that $(Y_t)_{t=0}^\infty$ is also a nonnegetive supermartingale with respect to the filtration $(\mathcal{F}_{\mathbf{x}_0,t})_{t=0}^\infty$. The nonnegativity part of the claim clearly holds since each $X_t$ is nonnegative. To check the expected decrease condition of supermartingales, for each $t \in \mathbb{N}_0$ and for each $\rho \in \Omega_{\mathbf{x_0}}$ we have distinguish two cases:

1. If $V(\mathbf{x}_i) < \lambda$ for all $0 \leq i \leq t$, then

$$\mathbb{E}_{\mathbf{x}_0}[Y_{t+1} \mid \mathcal{F}_{\mathbf{x}_0,t}](\rho)$$
$$= \mathbb{E}_{\mathbf{x}_0}[Y_{t+1} \cdot \mathbb{I}(V(\mathbf{x}_{t+1}) < \lambda) \mid \mathcal{F}_{\mathbf{x}_0,t}](\rho) + \mathbb{E}_{\mathbf{x}_0}[Y_{t+1} \cdot \mathbb{I}(V(\mathbf{x}_{t+1}) \geq \lambda) \mid \mathcal{F}_{\mathbf{x}_0,t}](\rho)$$
$$= \frac{1}{\gamma^{t+1}} \cdot \mathbb{E}_{\mathbf{x}_0}[X_{t+1} \cdot \mathbb{I}(V(\mathbf{x}_{t+1}) < \lambda) \mid \mathcal{F}_{\mathbf{x}_0,t}](\rho) + \mathbb{E}_{\mathbf{x}_0}[Y_t \cdot \mathbb{I}(V(\mathbf{x}_{t+1}) \geq \lambda) \mid \mathcal{F}_{\mathbf{x}_0,t}](\rho)$$
$$= \frac{1}{\gamma^{t+1}} \cdot \mathbb{E}_{\mathbf{x}_0}[X_{t+1} \mid \mathcal{F}_{\mathbf{x}_0,t}](\rho) - \frac{1}{\gamma^{t+1}} \cdot \mathbb{E}_{\mathbf{x}_0}[X_{t+1} \cdot \mathbb{I}(V(\mathbf{x}_{t+1}) \geq \lambda) \mid \mathcal{F}_{\mathbf{x}_0,t}](\rho)$$
$$+ \mathbb{E}_{\mathbf{x}_0}[Y_t \cdot \mathbb{I}(V(\mathbf{x}_{t+1}) \geq \lambda) \mid \mathcal{F}_{\mathbf{x}_0,t}](\rho)$$
$$\leq \frac{1}{\gamma^{t+1}} \cdot \gamma \cdot X_t(\rho) - \mathbb{E}_{\mathbf{x}_0}[(\frac{1}{\gamma^{t+1}} \cdot X_{t+1} - Y_t) \cdot \mathbb{I}(V(\mathbf{x}_{t+1}) \geq \lambda) \mid \mathcal{F}_{\mathbf{x}_0,t}](\rho)$$
$$= Y_t(\rho) - \mathbb{E}_{\mathbf{x}_0}[(\frac{1}{\gamma^{t+1}} \cdot X_{t+1} - Y_t) \cdot \mathbb{I}(V(\mathbf{x}_{t+1}) \geq \lambda) \mid \mathcal{F}_{\mathbf{x}_0,t}](\rho)$$
$$\leq Y_t(\rho).$$

   The first equality holds by the law of total probability. The second equality holds by the definition of $Y_{t+1}$. The third equality holds by the law of total probability. The fourth inequality holds since above we proved that $\mathbb{E}_{\mathbf{x}_0}[X_{t+1} \mid \mathcal{F}_{\mathbf{x}_0,t}](\rho) \leq \gamma \cdot X_t$ whenever $V(\mathbf{x}_i) < \lambda$ for all $0 \leq i \leq t$ (cases 1 and 2 above). The fifth equality holds by definition of $Y_t$ and the assumption that $V(\mathbf{x}_i) < \lambda$ for all $0 \leq i \leq t$. Finally, the sixth inequality follows by observing that in the case when $V(\mathbf{x}_i) < \lambda$ for all $0 \leq i \leq t$ but $V(\mathbf{x}_{t+1}) \geq \lambda$, we have $X_{t+1}/\gamma^{t+1} \geq \lambda/\gamma^{t+1} \geq \lambda/\gamma^t \geq Y_t$.
2. If $V(\mathbf{x}_i) = \lambda$ for some $0 \leq i \leq t$, then $\mathbb{E}_{\mathbf{x}_0}[Y_{t+1} \mid \mathcal{F}_{\mathbf{x}_0,t}](\rho) = Y_t(\rho) = Y_i(\rho)$.

Hence, we have proved that $(Y_t)_{t=0}^\infty$ is a nonnegative supermartingale.

We conclude the theorem claim by observing that

$$\mathbb{P}_{\mathbf{x}_0}\left[\sup_{t \geq 0} X_t < \lambda\right] = \mathbb{P}_{\mathbf{x}_0}\left[\sup_{t \geq 0} \gamma^t \cdot Y_t < \lambda\right] = \mathbb{P}_{\mathbf{x}_0}\left[\sup_{t \geq N} \gamma^t \cdot Y_t < \lambda\right]$$
$$= \mathbb{P}_{\mathbf{x}_0}\left[\gamma^N \cdot \sup_{t \geq N} \gamma^{t-N} \cdot Y_t < \lambda\right] = \mathbb{P}_{\mathbf{x}_0}\left[\sup_{t \geq N} \gamma^{t-N} \cdot Y_t < \frac{\lambda}{\gamma^N}\right]$$
$$\geq \mathbb{P}_{\mathbf{x}_0}\left[\sup_{t \geq N} Y_t < \frac{\lambda}{\gamma^N}\right] \geq \mathbb{P}_{\mathbf{x}_0}\left[\sup_{t \geq 0} Y_t < \frac{\lambda}{\gamma^N}\right] \geq 1 - \frac{1}{\lambda} \cdot \gamma^N.$$

Three non-trivial steps are the first and the second equality and the last inequality. The first equality holds since, if $\sup_{t \geq 0} X_t < \lambda$, then we also have $Y_t = X_t/\gamma^t$ for each $t$ by the definition of $Y_t$. The second equality holds since the system cannot reach a state in which $V \geq \lambda$ and so $X_t \geq \lambda$ in less than $N$ time steps. On the other hand, for the last inequality we the inequality in Theorem 7. Applying the inequality to $(Y_t)_{t=0}^\infty$ and $\frac{\lambda}{\gamma^N}$ and observing that $\mathbb{E}[Y_0] = \mathbb{E}[X_0] \leq 1$ by the Initial condition in Definition 2 yields $\mathbb{P}_{\mathbf{x}_0}[\sup_{t \geq 0} Y_t \geq \lambda/\gamma^N] \leq \frac{\gamma^N}{\lambda}$ and thus the last inequality. $\qquad\square$

## E   Learning Policies with Reach-avoid Supermartingales

We now present the POLICY+RASM subprocedure that we use for simultaneously learning a policy $\pi_\mu$ and an RASM $V_\theta$, both of which are parametrized as neural networks with parameters $\mu$ and $\theta$. The subprocedure POLICY+RASM is *identical* to the algorithm of [68], thus we keep this exposition brief and refer the reader to [68] for details. The reason why we can reuse this algorithm even though it learns additive RASMs is that additive and multiplicative RASMs are equivalent by Thereom 1. As we show below, the algorithm does not need to explicitly set an additive term $\epsilon$ or a multiplicative factor $\gamma$, thus it is applicable to learning both additive and multiplicative RASMs. We then show how to use $V_\theta$ to extract the bound in Theorem 2.

Analogously as in [68], the value $\lambda > 1$ in Definition 3 is an algorithm parameter and we initialize it to $\lambda = \frac{1}{1-p'}$ so that the Theorem 2 bound $1 - \frac{1}{\lambda} \cdot \gamma^N \geq 1 - \frac{1}{\lambda} = p'$ implies satisfaction of the desired probabilistic reach-avoid specification. If the algorithm succeeds in learning $\pi_\mu$ and $V_\theta$ with this value of $\lambda$, then the reach-avoid problem is solved. Otherwise, the algorithm gradually decreases the value of $\lambda$ and tries to relearn $\pi_\mu$ and $V_\theta$ so that the resulting bound in Theorem 2 still exceeds $p'$. Thus, our new bound also yields an improvement in the algorithm.

The algorithm consists of two modules called *learner* and *verifier*, which are composed into a loop. In each loop iteration, the learner first learns a policy $\pi_\mu$ and an RASM candidate $V_\theta$. These are then passed to the verifier which formally checks whether $V_\theta$ satisfies all conditions in Definition 2. If the verification is successful, the algorithm returns the policy. Otherwise, the verifier identifies *counterexample* states at which the additive RASM conditions are violated. These are then passed to the learner and are used to fine-tune the previously learned policy and RASM by refining the loss function using the computed counterexamples.

**Learner** A policy $\pi_\mu$ and an additive RASM candidate $V_\theta$ are learned by minimizing the loss function

$$\mathcal{L}(\theta, \nu) = \mathcal{L}_{\text{Init}}(\nu) + \mathcal{L}_{\text{Unsafe}}(\nu) + \mathcal{L}_{\text{Dec}}(\theta, \nu) + \mathcal{L}_{\text{Lipschitz}}(\theta, \nu).$$

The first three loss terms are constructed from the sets $C_{\text{init}}$, $C_{\text{unsafe}}$ and $C_{\text{dec}}$ which are initialized by computing finite discretizations of $\mathcal{X}_0$, $\mathcal{X}_u$ and $\mathcal{X} \backslash \mathcal{X}_t$ and are later extended by counterexamples computed by the verifier. The loss terms are used to guide the learner towards learning a true additive RASM which satisfies the Initial, Safety and Expected decrease conditions. Each loss term is designed to incur a loss at a counterexample whenever that counterexample violates the corresponding condition. In order for the Nonnegativity condition to be satisfied by default, the algorithm applies the softplus activation function to the output of $V_\theta$. The loss term $\mathcal{L}_{\text{Lipschitz}}(\theta, \nu)$ is a regularization term that does not enforce any of the defining conditions of additive RASMs, however it helps in decreasing the Lipschitz constants of neural networks. Each loss term is defined as follows:

$$\mathcal{L}_{\text{Init}}(\nu) = \max_{\mathbf{x} \in C_{\text{init}}} \{V_\nu(\mathbf{x}) - 1, 0\}$$

$$\mathcal{L}_{\text{Unsafe}}(\nu) = \max_{\mathbf{x} \in C_{\text{unsafe}}} \big\{ \frac{1}{1-p} - V_\nu(\mathbf{x}), 0 \big\}$$

$$\mathcal{L}_{\text{Decrease}}(\theta, \nu) = \frac{1}{|C_{\text{dec}}|} \cdot$$

$$\sum_{\mathbf{x} \in C_{\text{decrease}}} \Big( \max \Big\{ \sum_{\omega_1, \dots, \omega_N \sim \mathcal{N}} \frac{V_\nu\big(f(\mathbf{x}, \pi_\theta(\mathbf{x}), \omega_i)\big)}{N} - V_\theta(\mathbf{x}) + \tau \cdot K, 0 \Big\} \Big)$$

The last loss term $\mathcal{L}_{\text{Lipschitz}}(\theta, \nu) = t \cdot (\mathcal{L}_{\text{Lipschitz}}(\theta) + \mathcal{L}_{\text{Lipschitz}}(\nu))$ is the regularization term used to guide the learner towards learning neural networks whose Lipschitz constants are below a tolerable threshold $\rho$, where $t > 0$ is a regularization constant. By preferring networks with small Lipschitz constants, we allow the verifier to use a wider mesh and thus make verification condition easier to satisfy. We have The regularization term for $\pi_\theta$ (and analogously for $V_\nu$) is defined via

$$\mathcal{L}_{\text{Lipschitz}}(\theta) = \max \Big\{ \prod_{W, b \in \theta} \max_j \sum_i |W_{i,j}| - \rho, 0 \Big\},$$

where $W$ and $b$ weight matrices and bias vectors for each layer in $\pi_\theta$.

**Verifier** The verifier checks whether $V_\theta$ satisfies the defining properties of additive RASMs in Definition 2. Recall, the Nonnegativity condition is satisfied by default due to the softplus activation function applied to the output layer of $V_\theta$. Hence, the verifier only needs to check the Initial, Safety and Expected decrease conditions.

Since $f$, $\pi_\mu$ and $V_\theta$ are continuous functions defined over a compact domain $\mathcal{X}$ thus also Lipschitz continuous, the verifier may check the (both Multiplicative and Additive) expected decrease condition by checking a slightly stricter condition at finitely many discretization points. A *discretization* of the state space $\mathcal{X}$ with *mesh* $\tau > 0$ is a finite set $\tilde{\mathcal{X}} \subseteq \mathcal{X}$ such that, for every $\mathbf{x} \in \mathcal{X}$, there exists $\tilde{\mathbf{x}} \in \tilde{\mathcal{X}}$ such that $||\mathbf{x} - \tilde{\mathbf{x}}||_1 < \tau$. The discretization is computed by taking a grid of mesh $\tau$. Then, to check the expected decrease condition, it was showed in [68] that it suffices to check for each $\tilde{\mathbf{x}} \in \tilde{\mathcal{X}}$ whose

adjacent discretization grid cells contain a non-target state and over which $V$ attains a value that is less than or equal to $\lambda$ that

$$\mathbb{E}_{\omega \sim d}\left[V_\theta\Big(f(\tilde{\mathbf{x}}, \pi(\tilde{\mathbf{x}}), \omega)\Big)\right] < V_\theta(\tilde{\mathbf{x}}) - \tau \cdot K,$$

where $K = L_V \cdot (L_f \cdot (L_\pi + 1) + 1)$ and $L_f$, $L_\pi$ and $L_V$ are Lipschitz constants of $f$, $\pi_\mu$ and $V_\theta$. It is assumed that $L_f$ is provided and $L_\pi$ and $L_V$ are computed by the method of [57]. To verify the Initial condition, the verifier collects the set $\text{Cells}_{\mathcal{X}_0}$ of all cells of the discretization grid that intersect the initial set $\mathcal{X}_0$. Then, for each cell $\in \text{Cells}_{\mathcal{X}_0}$, it checks if $\sup_{\mathbf{x} \in \text{cell}} V_\theta(\mathbf{x}) \leq 1$, where the supremum is bounded from above by using interval arithmetic abstract interpretation (IA-AI) [22, 27] to propagate across neural network layers the extreme values that $V_\theta$ can attain over a cell. Similarly, to verify the Unsafe condition, the verifier collects the set $\text{Cells}_{\mathcal{X}_u}$ of all cells of the discretization grid that intersect the initial set $\mathcal{X}_u$. Then, for each cell $\in \text{Cells}_{\mathcal{X}_u}$, it uses IA-AI to check if $\inf_{\mathbf{x} \in \text{cell}} V_\theta(\mathbf{x}) \geq \lambda$.

If the verifier shows that $V_\theta$ satisfies the above checks, it concludes that $V_\theta$ is an additive (and therefore multiplicative) RASM for the system under the policy $\pi_\mu$ and returns the policy together with the lower bound on the probability of satisfying the reach-avoid specification as in Theorem 2. The fact that the verifier is correct was proved in [68, Theorem 2]. Otherwise, if a counterexample $\tilde{\mathbf{x}}$ to any of the checks is found, it is added to one of the three counterexample sets $C_{\text{init}}$, $C_{\text{unsafe}}$ and $C_{\text{dec}}$ that are then used by the learner to fine-tune $V_\theta$ and $\pi_\mu$.

**Soundness and computation of $\gamma$.** The following theorem establishes that the above is a sound verification procedure and provides a closed-form expression for the values of $\delta > 0$ and $\gamma \in (0, 1)$ for which $V_\theta$ is a $(\gamma, \delta, \lambda)$-multiplicative RASM. Hence, to compute the lower bound on the probability of satisfying reach-avoidance in Theorem 2, one may use the value of $\gamma$ implied by the theorem together with $\lambda$ which is fixed by the algorithm, $L_V$ which is computed by the algorithm and the maximal step size $\Delta$ which we assume is provided by the user.

**Theorem 8.** *If the verifier returns neural networks $\pi_\mu$ and $V_\theta$, then $V_\theta$ is a $(\gamma, \delta, \lambda)$-multiplicative RASM with*

$$\delta = \min\left\{ \min_{\tilde{\mathbf{x}} \in \tilde{\mathcal{X}}}\Big(V_\theta(\tilde{\mathbf{x}}) - \tau \cdot K - \mathbb{E}_{\omega \sim d}[V_\theta(f(\tilde{\mathbf{x}}, \pi(\tilde{\mathbf{x}}), \omega))]\Big), \lambda\right\}, \quad \gamma = 1 - \frac{\delta}{\lambda}.$$

*Proof.* Since it was shown in [68] that the verifier provides a sound verification procedure for checking that $V_\theta$ is an additive RASM, by the equivalence in Theorem 1 it follows that it is also sound for checking that $V_\theta$ is a $(\gamma, \delta, \lambda)$-multiplicative RASM for some values of $\gamma$ and $\delta$. Hence, it remains to show that the values of $\gamma$ and $\delta$ in the theorem statement are correct.

First, we show that the Strict positivity outside $\mathcal{X}_t$ condition is satisfied with the above value of $\delta$. To see this, let $\mathbf{x} \in \mathcal{X} \backslash \mathcal{X}_t$. Since $\delta \leq \lambda$, suppose without loss of generality that $V_\theta(\mathbf{x}) \leq \lambda$. Then, $\mathbf{x}$ is contained in a discretization grid cell which contains a non-target state and over which $V$ attains a value that is $\leq \lambda$. Hence, the verifier has shown that $\mathbb{E}_{\omega \sim d}[V_\theta(f(\tilde{\mathbf{x}}, \pi(\tilde{\mathbf{x}}), \omega))] < V_\theta(\tilde{\mathbf{x}}) - \tau \cdot K$ holds for each $\tilde{\mathbf{x}} \in \tilde{\mathcal{X}}$ which is a vertex of this cell. Taking a vertex $\tilde{\mathbf{x}} \in \tilde{\mathcal{X}}$ of this cell for which $||\mathbf{x} - \tilde{\mathbf{x}}||_1 \leq \tau$, by the definition of Lipscthiz constants we have

$$\begin{aligned}
&\mathbb{E}_{\omega \sim d}\left[V\Big(f(\mathbf{x}, \pi(\mathbf{x}), \omega)\Big)\right] \\
&\leq \mathbb{E}_{\omega \sim d}\left[V\Big(f(\tilde{\mathbf{x}}, \pi(\tilde{\mathbf{x}}), \omega)\Big)\right] + ||f(\tilde{\mathbf{x}}, \pi(\tilde{\mathbf{x}}), \omega) - f(\mathbf{x}, \pi(\mathbf{x}), \omega)||_1 \cdot L_V \\
&\leq \mathbb{E}_{\omega \sim d}\left[V\Big(f(\tilde{\mathbf{x}}, \pi(\tilde{\mathbf{x}}), \omega)\Big)\right] + ||(\tilde{\mathbf{x}}, \pi(\tilde{\mathbf{x}}), \omega) - (\mathbf{x}, \pi(\mathbf{x}), \omega)||_1 \cdot L_V \cdot L_f \\
&\leq \mathbb{E}_{\omega \sim d}\left[V\Big(f(\tilde{\mathbf{x}}, \pi(\tilde{\mathbf{x}}), \omega)\Big)\right] + ||\tilde{\mathbf{x}} - \mathbf{x}||_1 \cdot L_V \cdot L_f \cdot (1 + L_\pi) \\
&\leq \mathbb{E}_{\omega \sim d}\left[V\Big(f(\tilde{\mathbf{x}}, \pi(\tilde{\mathbf{x}}), \omega)\Big)\right] + \tau \cdot L_V \cdot L_f \cdot (1 + L_\pi).
\end{aligned}$$

Hence, by the Nonnegativity condition we also have

$$
\begin{aligned}
V(\mathbf{x}) &\geq V(\mathbf{x}) - \mathbb{E}_{\omega \sim d}\Big[V\Big(f(\mathbf{x}, \pi(\mathbf{x}), \omega)\Big)\Big] \\
&\geq V(\tilde{\mathbf{x}}) - \tau \cdot L_V - \mathbb{E}_{\omega \sim d}\Big[V\Big(f(\tilde{\mathbf{x}}, \pi(\tilde{\mathbf{x}}), \omega)\Big)\Big] - \tau \cdot L_V \cdot L_f \cdot (1 + L_\pi) \\
&= V(\tilde{\mathbf{x}}) - \tau \cdot K - \mathbb{E}_{\omega \sim d}\Big[V\Big(f(\tilde{\mathbf{x}}, \pi(\tilde{\mathbf{x}}), \omega)\Big)\Big] \\
&\geq \delta,
\end{aligned}
\tag{2}
$$

since $K = L_V \cdot (L_f \cdot (L_\pi + 1) + 1)$, which concludes the proof.

Second, we show that Multiplicative expected decrease condition with the above value of $\gamma$ holds. Let $\mathbf{x} \in \mathcal{X} \backslash \mathcal{X}_t$ be such that $V_\theta(\mathbf{x}) \leq \lambda$. We need to show that $\gamma \cdot V(\mathbf{x}) \geq \mathbb{E}_{\omega \sim d}[V(f(\mathbf{x}, \pi(\mathbf{x}), \omega))]$. Since we showed in eq. (2) that $V(\mathbf{x}) \geq V(\mathbf{x}) - \mathbb{E}_{\omega \sim d}[V(f(\mathbf{x}, \pi(\mathbf{x}), \omega))] \geq \delta > 0$ and since $V_\theta(\mathbf{x}) \leq \lambda$, we have

$$
\begin{aligned}
\frac{\mathbb{E}_{\omega \sim d}[V(f(\mathbf{x}, \pi(\mathbf{x}), \omega))]}{V(\mathbf{x})} &= 1 - \frac{V(\mathbf{x}) - \mathbb{E}_{\omega \sim d}[V(f(\mathbf{x}, \pi(\mathbf{x}), \omega))]}{V(\mathbf{x})} \\
&\leq 1 - \frac{\delta}{V(\mathbf{x})} \leq 1 - \frac{\delta}{\lambda}.
\end{aligned}
$$

This concludes the proof. $\qquad\square$

## F  Proof of Theorem 3

**Theorem 3.** *[Proof in Appendix F] Algorithm 1 is compositional, and if it outputs a policy $\pi$, then $\pi$ guarantees the probabilistic specification $(\phi, p)$.*

*Proof.* In order to prove that $\pi$ guarantees satisfaction of the probabilistic specification $(\phi, p)$, by Theorem 5 it suffices to show that $\pi$ satisfies abstract reachability for the abstract graph $G$ with probability at least $p$.

To prove abstract reachability for $G$ with probability at least $p$, we show that a random trajectory of the system under policy $\pi$ satisfies reach-avoid specifications of the edges along the finite path $s = v_{i_0}, v_{i_1}, \ldots, v_{i_k} = t$ exhibited above with probability at least $p$. To prove this, we proceed by induction on $0 \leq j \leq k$ to show that a random trajectory of the system under policy $\pi$ satisfies reach-avoid specifications of each edge along a prefix $s = v_{i_0}, v_{i_1}, \ldots, v_{i_j}$ of this path with probability at least $\mathrm{Prob}[v_{i_j}]$. Recall, $\mathrm{Prob}$ is the dictionary computed by Algorithm 1. Abstract reachability for $G$ with probability at least $p$ then follows if we set $j = k$, since $v_{j_k} = t$ and we must have $\mathrm{Prob}[t] \geq p$ for Algorithm 1 to output a policy (lines 15-17).

The base case $j = 0$ follows trivially since the system starts in the initial region $\beta(s) = \mathcal{X}_0$ and since $\mathrm{Prob}[v_{i_0}] = \mathrm{Prob}[s] = 1$ by line 6. For the inductive step, suppose that $0 \leq j \leq k - 1$ and that $\pi$ satisfies reach-avoid specifications of each edge along $s = v_{i_0}, v_{i_1}, \ldots, v_{i_j}$ with probability at least $\mathrm{Prob}[v_{i_j}]$. The claim for the prefix of length $j + 1$ then follows by our construction of the finite path $s = v_{i_0}, v_{i_1}, \ldots, v_{i_k} = t$, as it implies that Algorithm 1 has successfully learned an edge policy for the edge $(v_{i_j}, v_{i_{j+1}})$ that ensures satisfaction of the associated reach-avoid specification with probability at least $p_{(v_{i_j}, v_{i_{j+1}})}$ and for which $\mathrm{Prob}[v_{i_{j+1}}] = p_{(v_{i_j}, v_{i_{j+1}})} \cdot \mathrm{Prob}[v_{i_j}]$. Since the right-hand-side of this equality is a lower bound on the probability of $\pi$ satisfying reach-avoid specifications of each edge along $s = v_{i_0}, v_{i_1}, \ldots, v_{i_j}$ multiplied by a lower bound on the probability of it satisfying the reach-avoid specification of the edge $(v_{i_j}, v_{i_{j+1}})$, the claim follows. This concludes the proof by induction. $\qquad\square$

