# OpenReview forum: "Compositional Policy Learning in Stochastic Control Systems with Formal Guarantees"
_NeurIPS.cc/2023/Conference — NeurIPS 2023 poster_

### Official Review · Reviewer_6cWz · 2023-07-02

**Soundness:** 4 excellent
**Presentation:** 4 excellent
**Contribution:** 2 fair
**Rating:** 6
**Confidence:** 5

**Summary:**

This paper proposes a verifiable RL framework that learns a composition of NN policies for stochastic control systems, along with a formal supermartingale certificate for the safety probability of satisfying the reach-avoid specification. It decomposes the global reach-avoid task into a DAG with edges denoting subtasks which are solved by policy+RASM. Finally, it composes the low-level subtask's policy into a global safe policy with probability guarantees. The authors evaluate this framework in a relatively simple Stochastic Nine Room environment.

**Strengths:**

This paper is greatly written and easy to follow. The authors did a good job on the paper presentation. They study an important safe RL problem with step-wise safety chance constraint, which is suitable for safety-critical systems.

The proposed compositional safe RL framework is novel to me.

The proposed algorithm and approach are sound.

Control policy learning with formal guarantees in probability is significant.

**Weaknesses:**

1. My biggest concern is the scalability problem of this paper, which might originate from the DAG representation. If the global RL task is too complex, the graph might be too huge to handle.

2. Because of the scalability problem, the experiments in this paper look much simpler than other RL papers.

3. The authors may miss a few recent papers addressing a similar safe RL problem with chance constraints, as in

a)  Enforcing Hard Constraints with Soft Barriers: Safe Reinforcement Learning in Unknown Stochastic Environments,

as far as I can tell, this reference also considers continuous control problems with safety probability chance constraints, and its guarantees are also based on the supermartingale property, although the approaches are quite different.

**Questions:**

1. What are the relations between (either additive or multiplicative)  RASM and barrier certificate for stochastic (either continuous or discrete) systems?
The definition of the barrier certificate can be found in the following references.

b) Stochastic safety verification using barrier certificates.

c) A Barrier Function Approach to Finite-Time Stochastic System Verification and Control.

2. What's the complexity of the proposed algorithm? It should be feasible to conduct the big  O complexity analysis as the algorithm is built on top of topologic sorting, binary searching, etc.




**Limitations:**

This paper may face a scalability problem.

---

> ### Author Rebuttal · Authors · 2023-08-08
>
> We thank the reviewer for the valuable feedback. In what follows, we answer the two questions raised by the reviewer.
>
> **Relation between RASMs and stochastic barrier functions.** On the high level, the main difference between RASMs and stochastic barrier functions is that RASMs consider reach-avoid specifications, whereas stochastic barrier functions only consider safety specifications.
> However, if we are only interested in safety, additive and multiplicative RASMs reduce to stochastic barrier functions by letting $X_t = \emptyset$ and $\epsilon = 0$ in Definition 2 for additive RASMs, and $\gamma = 1$ in Definition 3 for multiplicative RASMs. Here, we are referring to discrete-time stochastic barrier functions defined in Prajna et al. “Stochastic safety verification using barrier certificates”, CDC 2004. This was discussed in Zikelic et al. “Learning Control Policies for Stochastic Systems with Reach-Avoid Guarantees”, AAAI 2023, who introduced additive RASMs.
>
> Our multiplicative RASMs in Definition 3 syntactically resemble exponential stochastic barrier functions defined in Santoyo et al. “A Barrier Function Approach to Finite-Time Stochastic System Verification and Control”, Automatica 2021. Exponential stochastic barrier functions also impose a multiplicative expected decrease condition. However, they consider finite time horizon systems, where the time horizon N is given and known a priori, and show that exponential barrier functions provide bounds on safety probability which are tighter by a factor which is exponential in time horizon N . However, as $N \rightarrow \infty$, their bound reduces to the bound of Prajna et al. In contrast, in our Theorem 2 we show that our multiplicative RASMs provide tighter bounds on safety (or more generally, reach-avoid) probability even in unbounded (i.e. indefinite) or infinite time horizon systems.
>
> **Computational complexity.** The worst-case complexity of our algorithm is $\mathcal{O}(|E| \cdot RA)$, where E is the number of edges in the abstract graph and RA is an upper bound on the computational complexity of learning and verifying a reach-avoid policy. This is because both the topological sort and the forward pass on the abstract graph can be done in time which is linear in the number of edges.
> The computational complexity of RA is $\mathcal{O}(I \cdot (L + V))$, where \
> $I$ = bound on the number of learner-verifier iterations, \
> $L$ = complexity of the learner, and \
> $V$ = complexity of the verifier. \
> In general, the learner-verifier procedure is not guaranteed to converge (in which case our algorithm does not output a policy), so we introduce a parameter $I$ which is the maximal number of learner-verifier loop iterations per edge.
> Given that we consider learning over continuous state spaces and given that our loss function is not convex, we cannot bound $L$ and also need to introduce a timeout parameter.
> Finally, we have $V = \mathcal{O}((D / \tau)^n \cdot N)$ where $D$ is the diameter of the state space, $\tau$ is the mesh of the discretization used by the verifier, $n$ is the dimension of the state space, and $N$ is the number of neurons in the policy and RASM neural networks. This is because, for each cell in the discretization grid, interval-arithmetic abstract interpretation can verify all RASM defining conditions in time linear in the size of the networks. On the other hand, in the worst case there are $\mathcal{O}((D / \tau)^n)$ cells in the discretization grid.
>
> This results in a final bound on computational complexity
> $\mathcal{O}(|E| \cdot I \cdot (L + (D / \tau)^n \cdot N))$
> with the notation above.
>
> We will incorporate the above discussions into the final version of the paper. We also thank the reviewer for pointing out the recent ICML 2023 paper. We will discuss the comparison to our work in the final version. In particular, this paper considers a model-free setting and jointly learns a policy and a barrier function-like certificate, however it does not provide guarantees on correctness of the learned certificate.

---

> > ### Comment · Reviewer_6cWz · 2023-08-16
> >
> > Thanks for the detailed responses. They have sufficiently addressed my questions.
> > I will keep my score for now due to the scalability weakness.

---

### Official Review · Reviewer_E1c8 · 2023-07-03

**Soundness:** 4 excellent
**Presentation:** 4 excellent
**Contribution:** 3 good
**Rating:** 7
**Confidence:** 4

**Summary:**

This paper introduces CLAPS (Compositional Learning for Probabilistic Specifications), a new method for learning a composition of neural network policies in stochastic environments, together with a formal certificate which guarantees that a reach-avoid specification over the policy's behavior is satisfied with the desired probability.
The proposed approach is evaluated empirically on a stochastic Nine Rooms environment.

**Strengths:**

1) Different to previous works discussed by the authors, the CLAPS method is applicable to stochastic control systems, that may be defined via non-linear dynamics functions.

2) The approach is compositional. Complex tasks are analysed and solved in terms of simple tasks, that are then combined in order to achieve the overall goal.

3) The literature is discussed in detail. The bibliography, at 61 items, is an excellent overview of current methods in safe RL.


**Weaknesses:**

1) A lot of material appears in the appendix. I did not check it, but the paper appears to be consistent.

3) This work builds on top of [61] as regards the use of supermartingales. However, the bounds obtained are stricter, which seems to justify the claims to novelty (together with the compositional approach).

Minor, p. 3: When discussing the equation for the stochastic feedback loop system, u_t is described as the control action even though no u_t appears in the equation. On the other hand, function \pi is not described. I understand that \pi: X -> U is the policy and \pi(x_t) = u_t.


**Questions:**

p. 4: When the authors say "if the probability of a random trajectory...", do they actually mean "if the probability of any trajectory..."

**Limitations:**

p. 9: The authors mention that "the systematic decomposition used in our algorithm has advantages over manual task decompositions". They might discuss these advantages further.

---

> ### Author Rebuttal · Authors · 2023-08-08
>
> We thank the reviewer for the valuable feedback. In what follows, we answer the question raised by the reviewer.
>
> Indeed, on p. 4 we mean “if the probability of any trajectory”. This sentence refers to trajectories sampled from the probability space of all trajectories starting in initial state $x_0$, where the probability space is defined by the Markov chain semantics of the system under a control policy. We will clarify this in the final version of the paper.
>
> We will also address the minor remark on p. 3 and write $\pi: X \rightarrow U$ and $u_t = \pi(x_t)$ when defining the dynamics. We will also further discuss the advantages of systematic decomposition on p. 9.

---

> > ### Comment · Reviewer_E1c8 · 2023-08-14
> >
> > I am happy with the rebuttal provided by the authors. Thanks!

---

### Official Review · Reviewer_kcWZ · 2023-07-05

**Soundness:** 3 good
**Presentation:** 3 good
**Contribution:** 2 fair
**Rating:** 4
**Confidence:** 4

**Summary:**

This paper introduces CLAPS, a compositional method designed for learning and verifying neural network policies in stochastic control systems. By considering control tasks with specifications expressed in the SPECTRL language, CLAPS decomposes the task into an abstract graph of reach-avoid tasks. It utilizes reach-avoid supermartingales to offer formal guarantees on the probability of reach-avoidance in each subtask. Additionally, the paper establishes proof demonstrating that RASMs (Reach-Avoid Supermartingales) provide a significantly more stringent lower bound on the probability of reach-avoidance compared to prior approaches. The experimental evaluation conducted in the Stochastic Nine Rooms environment showcases the ability of CLAPS to derive guarantees for global compositional policies.

**Strengths:**

+ Compositional Learning is an important research direction. The approach presented in this paper provides correctness guarantees for individual sub-policies that can be used to collectively ensure the correctness of the global policy, making the learning and verification approach valuable for applications where safety and reliability are critical. The ability to verify and validate each component of the policy offers a robust and trustworthy framework for developing complex control systems.
+ Taking inspiration from exponential barrier certificates developed in the control theory community, the paper introduces a conceptually similar concept to improve Reach-Avoid Supermartingales, which provides a more strict lower bound on the probability of reach-avoidance guarantees (compared with prior work [61]).
+ The main algorithm is easy to understand, follow, and implement.

**Weaknesses:**

The proposed approach falls short of advancing the state-of-the-art in compositional learning and verification. While Reach-Avoid Supermartingales in prior work makes sense for providing probabilistic correctness guarantees for infinite time horizon systems, the expectation for a global policy composed of sub-policies is that each sub-policy terminates within a finite time horizon. Simpler techniques such as statistical verification, which relies on drawing a large number of samples and employing concentration inequalities, can achieve high-probability correctness guarantees for finite horizons. Consequently, the paper lacks a compelling argument showcasing why their probabilistic verification approach surpasses a straightforward statistical verification approach for compositional policies. Strengthening the paper would involve demonstrating that the probabilistic verification approach indeed outperforms stochastic verification in practice.

In the specific context of the Stochastic Nine Rooms environment examined in this paper, it seems that the robot can reach the goal within a finite time horizon. Given the convergence of PPO sub-policies, it is reasonable to anticipate that stochastic verification methods can yield substantially higher probabilistic guarantees compared to the reported result of 33% in the paper's probabilistic verification approach.

**Questions:**

+ Can the proposed Reach-Avoid Supermartingales (RASMs) approach be formally characterized in relation to exponential barrier certificates?

+ Have you considered applying Statistical Verification of Learning-Based Cyber-Physical Systems (https://cpsl.pratt.duke.edu/sites/cpsl.pratt.duke.edu/files/docs/zarei_hscc20.pdf) to the Nine Rooms environments?

**Limitations:**

The paper would benefit from a more comprehensive discussion and extensive experimentation regarding the advantages of the probabilistic verification approach over stochastic verification within the context of compositional policy learning.

---

> ### Author Rebuttal · Authors · 2023-08-08
>
> We thank the reviewer for the valuable feedback. In what follows, we answer the two questions raised by the reviewer.
>
> **Comparison of RASMs and exponential barrier functions.** Stochastic barrier functions were introduced for proving probabilistic safety in stochastic dynamical systems, i.e. without the additional reachability condition of reach-avoid specs (Prajna et al. “Stochastic Safety Verification Using Barrier Certificates”, CDC 2004). If we are only interested in probabilistic safety, RASMs reduce to stochastic barrier functions by setting $X_t = \emptyset$ together with $\epsilon = 0$ in Definition 2 for additive RASMs, and $\gamma = 1$ in Definition 3 for multiplicative RASMs. This was discussed in Zikelic et al. “Learning Control Policies for Stochastic Systems with Reach-Avoid Guarantees”, AAAI 2023, who introduced additive RASMs.
>
> To the best of our knowledge, exponential barrier functions for stochastic dynamical systems have been considered only for finite time horizon systems in which the time horizon N is fixed and known a priori (Santoyo et al. “A Barrier Function Approach to Finite-Time Stochastic System Verification and Control”, Automatica 2021). Exponential barrier functions also impose a multiplicative expected decrease condition, similar to our multiplicative RASMs. This allows them to provide tighter bounds on safety probability compared to classical stochastic barrier functions, which are tighter by a factor which is exponential in the time horizon N (Theorem 2 in Santoyo et al.). However, as $N \rightarrow \infty$, their bound reduces to the bound of Prajna et al.
> In contrast, Theorem 2 in our paper shows that our multiplicative RASMs provide tighter bounds on safety (or more generally, reach-avoid) probability even in unbounded (i.e. indefinite) or infinite time horizon systems.
>
> **Comparison of CLAPS and statistical verification.** We clarify that, while satisfaction of a reach-avoid specification (and, more generally, any SpectRL specification) can be witnessed by a finite trace, one needs infinite traces to witness that a reach-avoid specification is not satisfied. For instance, in the Nine Rooms environment, one can design a policy under which a trace remains stuck in one room indefinitely without violating safety constraints and without reaching the target room. However, one cannot witness this by sampling finite traces, since any finite trace can also be extended to an infinite trace that eventually leaves the room.
> One way to overcome this limitation would be to sample traces of some fixed length and then treat all longer traces as either satisfying or violating the specification. However, such an approach would either be overestimating or underestimating the probability of a trace satisfying the specification and would not provide statistical guarantees. This means that statistical methods are applicable to and effective in finite time horizon systems, but they are not applicable to our setting. Our algorithm (CLAPS) provides formal guarantees even for unbounded (i.e. indefinite) or infinite time horizon systems. This is because the guarantees provided by RASMs do not impose any restrictions on the time horizon.
>
> We will incorporate the above discussions into the final version of the paper, as we believe they will strengthen the paper.

---

> > ### Comment · Reviewer_kcWZ · 2023-08-14
> > **Still confused about the experiment settings**
> >
> > I thank the authors' response. However, I am still confused about the experiment settings and the argument made in the rebuttal. In this paper
> >
> > > Our method learns a policy along with a formal certificate which guarantees that a specification is satisfied with the desired probability.
> >
> > In the rebuttal
> >
> > > We clarify that, while satisfaction of a reach-avoid specification (and, more generally, any SpectRL specification) can be witnessed by a finite trace, one needs infinite traces to witness that a reach-avoid specification is not satisfied.
> >
> > Does your argument retain its validity if I'm only concerned about verifying whether a specification is **satisfied** with a certain desired probability?
> >
> > I concur with your assessment that your work has strength in providing probabilistic correctness assurances for systems with infinite time horizons, such as verifying the balance of a pendulum. However, in the context of a global policy composed of sub-policies, it's expected that each sub-policy concludes within a finite time horizon. Expecting a sub-policy to run infinitely seems unreasonable. Upon reaching a sub-goal, why not transition to the subsequent sub-policy for the next sub-goal? What's the rationale behind verifying a sub-policy that **satisfies** its specification with the desired probability over an infinite time horizon?
> >
> > While statistical methods are only suited for and effective in finite time horizon systems, they seamlessly align with your scenario if your objective is to ensure that a specification is **satisfied** with the desired probability, and each sub-policy is anticipated to reach a known sub-goal. Even if traces longer than a certain threshold are treated as violating a specific reach-avoid specification, stochastic verification methods could conceivably provide significantly higher probabilistic assurances than the reported 33% result. In this context, why can this experimental outcome be used to validate the effectiveness of your approach?

---

> > > ### Author Response · Authors · 2023-08-19
> > > **Response**
> > >
> > > We thank the reviewer for the response.
> > >
> > > The key assumption required to use statistical verification methods is that a finite threshold is known a-priori. As mentioned in the scenario, it is conceivable that this shortcoming can be side-stepped by stipulating that ‘traces longer than a certain threshold are treated as violating a specific reach-avoid specification’. However, in applications where we cannot make an accurate guess for such a threshold we would have no way to distinguish between bad (low satisfaction probability) sub-policies and a bad threshold guess. For such situations it would be useful to have methods that provide formal guarantees without making an assumption about the finite threshold.
> > >
> > > As a consequence of this assumption CLAPS should be experimentally compared to other formal guarantee methods, where we provide better results (in particular the comparison of multiplicative and additive RASMs in Table 1). Additionally, the satisfaction bound for CLAPS (Theorem 2) does not depend on the number of environment interactions, which further highlights its applicability for situations when the satisfaction threshold might be arbitrarily long.
> > >
> > > We agree that an additional discussion about the benefits and drawbacks compared to statistical methods should be included in the paper and will add this to the final version. Including a clear statement about how statistical methods are likely to provide higher probabilistic assurances when the threshold is guessed correctly but might fail if the threshold is under-approximated. Our view is that statistical and formal methods should be viewed as complementary, where statistical methods give an estimate of the satisfaction of the reach-avoid spec, whereas we provide a lower bound​​.

---

> > > > ### Comment · Reviewer_kcWZ · 2023-08-20
> > > > **Thanks for your response**
> > > >
> > > > **Making an assumption about the finite threshold:** This seems puzzling. To train the sub-policy, one needs to define the maximum time steps allowed, a requirement inherent in any practical RL algorithm. Given this parameter, why would guessing a finite threshold be necessary?

---

> > > > > ### Comment · Reviewer_kcWZ · 2023-08-20
> > > > >
> > > > > One potential method to highlight the advantage of this approach over statistical verification is to examine larger room layouts, where trained sub-policies are used in environments distinct from those in which they were trained, making guessing the finite threshold necessary.
> > > > >
> > > > > In summary, it looks like the paper lacks experiments that illustrate the notion that "statistical and formal methods should be viewed as complementary" within the framework of compositional verification of sub-policies. I consider this comparison crucial, and as such, I maintain my belief that the paper falls below the threshold for acceptance.

---

> > > > > > ### Author Response · Authors · 2023-08-21
> > > > > >
> > > > > > Respectfully, we disagree with the assertion that “[O]ne needs to define the maximum time steps allowed, a requirement inherent in any practical RL algorithm”. There is substantial theoretical and empirical work that has established that RL algorithms can work in the infinite-horizon setting without this assumption [1, 2, 3, 4, 5]. These techniques usually make some assumption about what is known about the underlying MDP, however these assumptions are not directly comparable to a finite-horizon assumption.
> > > > > >
> > > > > > Certainly the assumption of a finite-horizon makes it easier to train a performant RL algorithm for certain tasks. However, infinite-horizon RL algorithms have been making significant and steady progress over the years, and there are real-world applications where the finite-horizon assumption is too restrictive. It is noteworthy that this assumption is essential for statistical verification techniques, and to the best of our knowledge there are no methods that work without it (the discussion in the last paragraph of Section 10 in this survey [6] agrees with this view stating, “In addition, most of the proposed SMC results are suitable for finite-time horizons. More precisely, in the setting of SMC approaches in infinite-time horizons, the proposed results require some strong assumptions that are not in general satisfiable by SHS.”).
> > > > > >
> > > > > > Given this known fundamental limitation, an experimental comparison does not add any useful information because we know for certain what the outcome will be, i.e. for an under-approximated finite threshold the statistical method will completely fail to verify all policies (while finding better assurances when a good threshold can be guessed). The fact that current statistical techniques provide better assurances with the finite-horizon (or good threshold) assumption might erroneously suggest that methods building on formal verification techniques are not useful. We maintain that experimental comparisons are meaningful if made between techniques with comparable underlying assumptions and nature of guarantees.
> > > > > >
> > > > > > [1] Wei et al. 2020. Model-free Reinforcement Learning in Infinite-horizon Average-reward Markov Decision Processes. ICML.
> > > > > > [2] Wang et al. 2020. Q-learning with UCB Exploration is Sample Efficient for Infinite-Horizon MDP. ICML.
> > > > > > [3] Sidford et al. 2017. Variance Reduced Value Iteration and Faster Algorithms for Solving Markov Decision Processes. Naval Research Logistics.
> > > > > > [4] Li et al. 2022.  Infinite-Horizon Reach-Avoid Zero-Sum Games via Deep Reinforcement Learning.
> > > > > > [5] Hsu et al. 2021. Safety and Liveness Guarantees through Reach-Avoid Reinforcement Learning. Robotics: Science and Systems.
> > > > > > [6] Lavaei et al. Automated verification and synthesis of stochastic hybrid systems: A survey.  Automatica.

---

> > > > > > > ### Comment · Reviewer_kcWZ · 2023-08-21
> > > > > > >
> > > > > > > Thank you for the clarification. I withdraw my comment on "any practical RL algorithms" as I understand its subjectivity. I want to reiterate my appreciation for the value of prior work [61], which provides guarantees over an infinite time horizon, a critical and valuable contribution. My confusion for this submission arises from the compositional verification of sub-policies in a context where sub-policies are assumed to run infinitely, while your implementation must have imposed a definite time limit on their execution (since you used PPO).
> > > > > > >
> > > > > > > Stepping back, let's consider the scenario where the finite threshold for sub-policy execution steps must be inferred. Even using statistical verification, I can optimize this threshold through a simple line search, which might yield better results than those reported.
> > > > > > >
> > > > > > > What I intend to highlight is a potential flaw in the problem setting. The central issue is that your paper attempts to address an infinite time horizon verification problem but assumes the abstract graph from reach-avoid specifications is acyclic. To truly showcase the advantages of infinite time horizon verification, the assumption of an acyclic abstract graph might need to be reconsidered. This could involve verifying systems that genuinely operate infinitely, calling sub-policies repeatedly in a cycle to satisfy specifications that describe sets
> > > > > > > (languages) of infinite words (or, $\omega$-languages).

---

> > > > > > > > ### Author Response · Authors · 2023-08-21
> > > > > > > >
> > > > > > > > It seems that there are two distinct points being raised here and we would like to address them separately:
> > > > > > > >
> > > > > > > > 1.Modify the problem setting because, “To truly showcase the advantages of infinite time horizon verification, the assumption of an acyclic abstract graph might need to be reconsidered.”
> > > > > > > >
> > > > > > > > Dropping the assumption of an acyclic abstract graph is a strictly harder problem, and this paper is not making claims of solving that. Many previous works in this area [27, 30] also assume an acyclic abstract graph. The infinite-time horizon problem can still manifest itself in the training of sub-policies and that would require the use of RL algorithms that work without an explicit assumption about the horizon. While the current implementation uses PPO, changing the training mechanism of sub-policies is a drop-in replacement. Additionally, we have shown that the bounds are not dependent on the number of environment interactions (Theorem 2) so our method will remain applicable for policy leaning techniques that specifically target the infinite-horizon setting. The experimental environments we are using are standard choices from the literature.
> > > > > > > >
> > > > > > > > 2. Statistical verification can provide better assurances even if we must learn the threshold (“Even using statistical verification, I can optimize this threshold through a simple line search, which might yield better results than those reported.”)
> > > > > > > >
> > > > > > > > We agree with the view that treating the threshold as a hyper-parameter might yield good statistical assurances in certain environments. However, to the best of our knowledge, existing techniques have not explored this option and there are no known bounds about the number of environment interactions that would be required to get a reasonable confidence in the learnt statistical guarantee. In fact, it is not even clear how to determine if the algorithm should continue searching for a better threshold or if it should start searching for a new policy. This is exactly why we feel that an experimental comparison is not meaningful here. A fair experimental comparison can only be made to existing techniques that make comparable underlying assumptions about the MDP.

---

> > > > > > > > > ### Comment · Reviewer_kcWZ · 2023-08-21
> > > > > > > > >
> > > > > > > > > Thanks for the response. For your item 1, I agree that prior work [27,30] explores acyclic abstract graphs. However, they do not claim they support infinite time horizons. For your item 2, please take a look at the following paper.
> > > > > > > > >
> > > > > > > > > Shuo Li, Osbert Bastani. Robust Model Predictive Shielding for Safe Reinforcement Learning with Stochastic Dynamics. ICRA 2020. (My apologies for incorrectly citing an irrelevant paper in the original comment)

---

> > > > > > > > > > ### Author Response · Authors · 2023-08-21
> > > > > > > > > >
> > > > > > > > > > Thank you for the additional citation. We will go through this carefully, however since the discussion period is coming to a close we would like to highlight that it seems the finite-horizon assumption (presented as bounded-time) is being made here as well. The last line of the first paragraph of Section 5 states: “For a PUS, verifying bounded-time properties is straightforward; verifying unbounded-time properties is more involving, and will be part of the future work.”
> > > > > > > > > >
> > > > > > > > > > Regarding the presentation in [27] and [30], many of their theoretical results are presented for the infinite-time horizon case (expressed as infinite trajectories), for example Theorem 3.4 in [30].

---

> > > > > > > > > > > ### Comment · Reviewer_kcWZ · 2023-08-21
> > > > > > > > > > >
> > > > > > > > > > > My apologies for incorrectly citing an irrelevant paper in my last comment.

---

> > > > > > > > > > > > ### Comment · Reviewer_kcWZ · 2023-08-21
> > > > > > > > > > > >
> > > > > > > > > > > > I read the paper [30] again. It seems the formalization there is in line with that of this paper. So I will not go against accepting this paper although some of my concerns remain. Thanks for the discussion.

---

> > > > > > > > > > > > > ### Author Response · Authors · 2023-08-21
> > > > > > > > > > > > >
> > > > > > > > > > > > > Thank you for the thoughtful discussion.

---

### Official Review · Reviewer_Dssn · 2023-07-06

**Soundness:** 2 fair
**Presentation:** 3 good
**Contribution:** 2 fair
**Rating:** 5
**Confidence:** 4

**Summary:**

The paper presents CLAPS, a compositional RL algorithm that also ensures guarantees of correctness when learning from temporal specifications. The core contribution of this work revolves around guarantees.

Prior works in compositional RL with guarantees apply to deterministic environments and/or ones with linear dynamics. This work presents an algorithm that is suitable for stochastic environments with non-linear dynamics. The algorithm builds on two prior works., namely [61] that learns policies for control systems with non-linear dynamics with reach-avoid guarantees and [30] that compositionally learns policies (without guarantees) from temporal specifications. An empirical evaluation on 9-rooms environment over a suite of source and target states shows the efficacy of the approach in learning long-horizon tasks with guarantees.

**Strengths:**

RL from temporal logic to learn long-horizon tasks is a promising and thriving research community. This paper makes a strong contribution in that space by learning policies with guarantees in stochastic systems with non-linear dynamics.

**Weaknesses:**

The empirical evaluation can be strengthened. Please find concrete comments below:

1. The evaluations have been presented for one environment only. The 9-room environment is somewhat low-dimensional. I am curious about how the approach may scale to higher dimensional environments.
2. Is there a comparison between the sample complexity of CLAPS with prior compositional approaches such as DiRL? This could offer a study of tradeoffs for guarantees.
3. Would it be possible to compare CLAPS to [27] on the benchmarks that would be common to both? It would be interesting to see how the formal guarantees of both approaches compare.

**Questions:**

I am quite unclear as to how the guarantee extends compositionally. From my understanding, a critical challenge in obtaining guarantees from compositional approaches such as [27] is that one needs to ensure the guarantee at the transition point between two edges, ie., in the region where one policy ends and the other begins. Could you clarify how that is accounted for in the proof of Theorem 5?

The example I have in mind is as follows: Say, one of the subgoal regions is split by a wall such that a learn policy entering the subgoal region ends before the wall and the policy exiting the subgoal region exits from the other side of the wall. The probability of connecting these two policies is clearly not 1. How is the guarantee in CLAPS accounting for such scenarios?



**Limitations:**

See above. I can change my score based on the clarification of the question above.

---

> ### Author Rebuttal · Authors · 2023-08-08
>
> We thank the reviewer for the valuable feedback. In what follows, we answer the questions raised by the reviewer.
>
> We clarify that Theorem 5 in the Appendix only states that a trajectory satisfies a SpectRL specification if and only if it satisfies abstract reachability in the abstract graph associated with the specification. Therefore, the probability of a SpectRL specification being satisfied is equal to the probability of abstract reachability being satisfied. However, Theorem 5 does not consider composition of reach-avoid guarantees associated to two edges.
>
> The composition guarantees are obtained by our composition of edge policies described in l.321-343 and proved correct in Theorem 3.  For each edge $(v_1,v_2)$ in the abstract graph, CLAPS synthesises an edge policy $\pi_{v_1,v_2}$ together with a lower bound $p_{v_1,v_2}$ on the probability of satisfying the reach-avoid specification associated to the edge (line 10 in Algorithm 1). The lower bound $p_{v_1,v_2}$ is the worst-case lower bound for *any initial state* in the region associated with the source vertex $v_1$.
> Hence, when composing guarantees of edges $(v_0, v_1)$ and $(v_1, v_2)$, CLAPS considers the worst-case state in the region associated to $v_1$ in which the agent may end up upon executing the policy associated to $(v_0,v_1)$, before moving on to the policy associated to $(v_1,v_2)$. Hence, the lower bound computed by CLAPS on the probability of satisfying the specification obtained by composing edges is $p_{v_0,v_1} \cdot p_{v_1,v_2}$. Our composition described in l.321-343 and proved correct in Theorem 3 formalises this reasoning and ensures that the guarantees obtained by composing edge policies are indeed correct. We will clarify this further in the final version of the paper.
>
> In the example described by the reviewer, this means that CLAPS would compute (1) a lower bound $p_{v_0,v_1}$ on the probability of satisfying the reach-avoid specification of the first edge, (2) a lower bound $p_{v_1,v_2}$ on the probability of satisfying the reach-avoid specification of the second edge (which would be attained in a state before the wall), and would conclude the lower bound of  $p_{v_0,v_1} \cdot p_{v_1,v_2}$ of satisfying the composition of two reach-avoid specifications.

---

### Decision · Program_Chairs · 2023-09-21

**Decision:**

Accept (poster)

**Comment:**

The authors develop a novel framework for verifying properties of a policy given properties of subpolicies that are composed to arrive at the global policy. The approach enables compositional verification of temporal specifications for RL agents, and constitutes a valuable contribution to the literature. Reviewer kcWZ had a thorough and respectful discussion with the authors, and having read through it, I conclude that while the reviewer still leans towards rejection, they do not have any technical objections on the paper. Furthermore, other reviewers favor acceptance. Hence I recommend acceptance.